# Whitening Convergence Rate of Coupling-based Normalizing Flows

**Felix Draxler**
Heidelberg University
felix.draxler@iwr.uni-heidelberg.de

**Christoph Schnörr**
Heidelberg University
schnoerr@math.uni-heidelberg.de

**Ullrich Köthe**
Heidelberg University
ullrich.koethe@iwr.uni-heidelberg.de

## Abstract

Coupling-based normalizing flows (e.g. RealNVP) are a popular family of normalizing flow architectures that work surprisingly well in practice. This calls for theoretical understanding. Existing work shows that such flows *weakly* converge to arbitrary data distributions [1]. However, they make no statement about the stricter convergence criterion used in practice, the maximum likelihood loss. For the first time, we make a quantitative statement about this kind of convergence: We prove that all coupling-based normalizing flows perform whitening of the data distribution (i.e. diagonalize the covariance matrix) and derive corresponding convergence bounds that show a linear convergence rate in the depth of the flow. Numerical experiments demonstrate the implications of our theory and point at open questions.

## 1 Introduction

Normalizing flows [2, 3] are among the most promising approaches to *generative* machine learning and have already demonstrated convincing performance in a wide variety of practical applications, ranging from image analysis [4, 5, 6, 7, 8] to astrophysics [9], mechanical engineering [10], causality [11], computational biology [12] and medicine [13]. As the name suggests, normalizing flows represent complex data distributions as bijective transformations (also known as flows or *pushforwards*) of standard normal or other well-understood distributions.

In this paper, we focus on a theoretical underpinning of coupling-based normalizing flows, a particularly effective class of normalizing flows in terms of invertible neural networks. All of the above applications are actually implemented using coupling-based normalizing flows. Their central building blocks are *coupling layers*, which decompose the space into two subspaces called *active* and *passive* subspace (see Section 3). Only the active dimensions are transformed conditioned on the passive dimensions, which makes the mapping computationally easy to invert. In order to vary the assignment of dimensions to the active and passive subspaces, coupling layers are combined with preceding orthonormal transformation layers into *coupling blocks*. These blocks are arranged into deep networks such that the orthonormal transformations are sampled uniformly at random from the orthogonal matrices and the coupling layers are trained with the maximum likelihood objective, see Equation (2). Upon convergence of the training, the sequence of coupling blocks gradually transforms the probability density that generated the given training data, into a standard normal distribution and vice versa.

36th Conference on Neural Information Processing Systems (NeurIPS 2022).

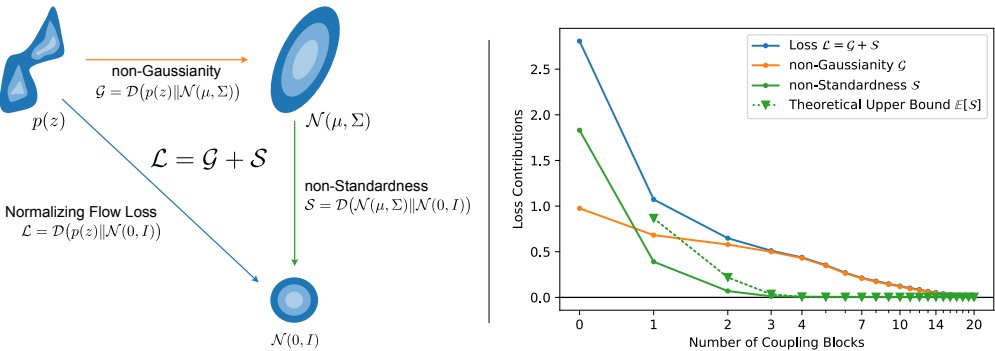

Figure 1: *(Left)* The Maximum Likelihood Loss $\mathcal{L}$ (blue) can be split into the *non-Gaussianity* $\mathcal{G}$ (orange) [25] and the *non-Standardness* $\mathcal{S}$ (green) of the latent code $z = f_\theta(x)$: $\mathcal{L} = \mathcal{G} + \mathcal{S}$ (Proposition 1). For the latter, we give explicit guarantees as one more coupling block is added in Theorems 1 and 2 and show a global convergence rate in Theorem 3. *(Right)* Typical fit of EMNIST digits by a standard affine coupling flow for various depths. Our theory (Theorem 1) upper bounds the average $\mathcal{S}$ for $L + 1$ coupling blocks given a trained model with $L$ coupling blocks (dotted green). We observe that our bound is predictive for how much end-to-end training reduces $\mathcal{S}$.

Since the resulting normalizing flows deviate significantly from *optimal* transport flows [14] and the bulk of the mathematical literature is focusing on optimal transport, an analysis tailored to coupling architectures is lacking. In a landmark paper, [1] proved that sufficiently large affine coupling flows weakly converge to arbitrary data densities. The notion of weak convergence is critical here, as *it does not imply convergence in maximum likelihood* [15, Remark 3]. Maximum likelihood (or, equivalently, the Kullback-Leibler (KL) divergence) is the loss that is actually used in practice. It can be used for gradient descent and it guarantees not only convergence in samples ("$x \sim q(x) \to x \sim p(x)$") but also in density estimates ("$q(x) \to p(x)$"). It is strong in the sense that the square root of the KL divergence upper bounds (up to a factor 2) the total variation metric, and hence also the Wasserstein metric if the underlying space is bounded [16]. Moreover, convergence under the KL divergence implies weak convergence which is fundamental for robust statistics [17].

We take a first step towards showing that coupling blocks also converge in terms of maximum likelihood. To the best of our knowledge, our paper presents for the first time a quantitative convergence analysis of coupling-based normalizing flows based on this strong notion of convergence.

Specifically, we make the following contributions towards this goal:

- We utilize that the loss of a normalizing flow can be decomposed into two parts (Figure 1): The divergence to the nearest Gaussian (*non-Gaussianity*) plus the divergence of that Gaussian to the standard normal (*non-Standardness*).

- The contribution of a single coupling layer on the non-Standardness is analyzed in terms of matrix operations (Schur complement and scaling).

- Explicit bounds for the non-Standardness after a single coupling block in expectation over all orthonormal transformations are derived.

- We use these results to prove that a sequence of coupling blocks whitens the data covariance and to derive linear convergence rates for this process.

Our results hold for all coupling architectures we are aware of (Appendix C), including: NICE [4], RealNVP [5], and GLOW [6]; Flow++ [18]; nonlinear-squared flow [19]; linear, quadratic [20], cubic [21], and rational quadratic splines [22]; neural autoregressive flows [23], and unconstrained monotonic neural networks [24]. We confirm our theoretical findings experimentally and identify directions for further improvement.

## 2 Related work

Analyzing which distributions coupling-based normalizing flows can approximate is an active area of research. A general statement shows that a coupling-based normalizing flow which can approximate an arbitrary invertible function can learn any probability density *weakly* [1]. This applies to affine coupling flows [4, 5, 6], Flow++ [18], neural autoregressive flows [26], and SOS polynomial flows [27]. Affine coupling flows converge to arbitrary densities in Wasserstein distance [15]. Both universality results, however, require that the couplings become ill-conditioned (i.e. the learnt functions become increasingly discontinuous as the error decreases, whereas in practice one observes that functions remain smooth). Also, they consider only a finite subspace of the data space. Even more importantly, the convergence criterion employed in their proofs (weak convergence resp. convergence under Wasserstein metric) is critical: Those criteria do not imply convergence in the loss that is employed in practice [15, Remark 3], the Kullback-Leibler divergence (equivalent to maximum likelihood). An arbitrarily small distance in any of the above metrics can even result in an infinite KL divergence. In contrast to previous work on affine coupling flows, we work directly on the KL divergence. We decompose it in two contributions and show the flow's convergence for one of the parts.

Regarding when ill-conditioned flows need to arise to fit a distribution, [28] showed that well-conditioned affine couplings can approximate log-concave padded distributions, again in terms of Wasserstein distance. Lipschitz flows on the other hand cannot model arbitrary tail behavior, but this can be fixed by adapting the latent distribution [29].

SOS polynomial flows converge in *total variation* to arbitrary probability densities [30], which also does not imply convergence in KL divergence; zero-padded affine coupling flows converge weakly [23], and so do Neural ODEs [31, 32]. Normalizing Flows with diagonal Jacobian in each layer, known as Gaussianization (Flows), also converge weakly [33, 34].

Closely related to our work, 48 linear affine coupling blocks can represent any invertible linear function $Ax + b$ with $\det(A) > 0$ [15, Theorem 2]. This also allows mapping any Gaussian distribution $\mathcal{N}(m, \Sigma)$ to the standard normal $\mathcal{N}(0, I)$. We put this statement into context in terms of the KL divergence: The loss is exactly composed of the divergence to the nearest Gaussian and of that Gaussian to the standard normal. We then make strong statements about the convergence of the latter, concluding that for typical flows a smaller number of layers is required for accurate approximation than predicted by [15].

## 3 Coupling-based normalizing flows

Normalizing flows learn an invertible function $f_\theta(x)$ that maps samples $x$ from some unknown distribution $p(x)$ given by samples to *latent variables* $z = f_\theta(x)$ so that $z$ follow a simple distribution, typically the standard normal. The function $f_\theta$ then yields an estimate $q(x)$ for the true data distribution $p(x)$ via the change of variables formula (e.g. [5]):

$$q(x) = \mathcal{N}(f_\theta(x); 0, I)|\det J|, \tag{1}$$

where $J = \nabla f_\theta(x)$ is the Jacobian of $f_\theta(x)$. We can train a normalizing flow via the maximum likelihood loss, which is equivalent to minimizing the Kullback-Leibler divergence between the distribution of the latent code $q(z)$, as given by $z = f_\theta(x)$ when $x \sim p(x)$, and the standard normal:

$$L = \mathcal{D}_{\mathrm{KL}}(q(z)\|\mathcal{N}(0, I)) = \mathbb{E}_{x\sim p(x)}\left[\tfrac{1}{2}\big\|f_\theta(x)\big\|^2 - \log|\det J|\right] + \mathrm{const}. \tag{2}$$

The invertible architecture that makes up $f_\theta$ has to (i) be computationally easy to invert, (ii) be able to represent complex transformations, and (iii) have a tractable Jacobian determinant $|\det J|$ [9]. Building such an architecture is an active area of research, see e.g. [2] for a review. In this work, we focus on the family of coupling-based normalizing flows, first presented in the form of the NICE architecture [4]. It is a deep architecture that consists of several blocks, each containing a rotation, a coupling and an ActNorm layer [6]:

$$f_{\mathrm{block}}(x) = (f_{\mathrm{act}} \circ f_{\mathrm{cpl}} \circ f_{\mathrm{rot}})(x). \tag{3}$$

The coupling $f_{\mathrm{cpl}}$ splits an incoming vector $x_0$ in two parts along the coordinate axis: The first part $p_0$, which we call *passive*, is left unchanged. The second part $a_0$, which we call *active*, is modified as

a function of the passive dimensions:

$$f_{cpl}(x_0) = f_{cpl}\begin{pmatrix} p_0 \\ a_0 \end{pmatrix} = \begin{pmatrix} p_0 \\ c(a_0; p_0) \end{pmatrix} =: \begin{pmatrix} p_1 \\ a_1 \end{pmatrix}. \tag{4}$$

Here, the coupling function $c : \mathbb{R}^{D/2} \times \mathbb{R}^{D/2} \to \mathbb{R}^{D/2}$ has to be a function that is easy to invert when $p_0$ is given, i.e. it is easy to compute $a_0 = c^{-1}(a_1; p_0)$ given $p_0$. This makes the coupling easy to invert: Call $x_1 = (p_1; a_1)$ the output of the layer, then $p_0 = p_1$. Use this to invert $a_1 = c(a_0; p_0)$. For example, RealNVP [5] proposes a simple affine transformation for $c$: $a_1 = c(a_0; p_0) = a_0 \odot s(p_0) + t(p_0)$ ($\odot$ means element-wise multiplication). $s(p_0) \in \mathbb{R}_+^{D/2}$ and $t(p_0) \in \mathbb{R}^{D/2}$ are computed by a feed-forward neural network. The coupling functions $c$ of other architectures our theory applies to are listed in Appendix C.

An Activation Normalization (ActNorm) layer [6] helps stabilize training and is implemented in practice like in the popular INN framework `FrEIA` [35]. It rescales and shifts each dimension:

$$f_{act}(x) = r \odot x + u, \tag{5}$$

given parameters $r \in \mathbb{R}_+^D$ and $u \in \mathbb{R}^D$. We include it as it simplifies our mathematical arguments.

If we were to concatenate several coupling layers, the entire network would never change the passive dimensions apart from the element-wise affine transformation in the ActNorm layer. Here, the rotation layers $f_{rot}(x) = Qx$ come into play [6]. They multiply an orthogonal matrix $Q$ to the data, changing which subspaces are passive respectively active. This matrix is typically fixed at random at initialization and then left unchanged during training.

## 4   Coupling layers as whitening transformation

The central mathematical question we answer in this work is the following: How can a deep coupling-based normalizing flow *whiten* the data? As the latent distribution is a standard normal, whitening is a necessary condition for the flow to converge. This is a direct property of the loss:

**Proposition 1** (Pythagorean Identity, Proof in Appendix B.1)*. Given data with distribution $p(x)$ with mean $m$ and covariance $\Sigma$. Then, the Kullback-Leibler divergence to a standard normal distribution decomposes as follows:*

$$\mathcal{D}_{KL}(p(x)\|\mathcal{N}(0,I)) = \underbrace{\mathcal{D}_{KL}(p(x)\|\mathcal{N}(m,\Sigma))}_{\textit{non-Gaussianity } \mathcal{G}(p)} + \underbrace{\mathcal{D}_{KL}(\mathcal{N}(m,\Sigma)\|\mathcal{N}(0,I))}_{\textit{non-Standardness } \mathcal{S}(p)}, \tag{6}$$

*and the non-Standardness again decomposes:*

$$\mathcal{S}(p) = \underbrace{\mathcal{D}_{KL}(\mathcal{N}(m,\Sigma)\|\mathcal{N}(m,\mathrm{Diag}(\Sigma)))}_{\textit{Correlation } \mathcal{C}(p)} + \underbrace{\mathcal{D}_{KL}(\mathcal{N}(m,\mathrm{Diag}(\Sigma))\|\mathcal{N}(0,I))}_{\textit{Diagonal non-Standardness}}. \tag{7}$$

This splits the transport from the data distribution to the latent standard normal into three parts: (i) From the data to the nearest Gaussian distribution $\mathcal{N}(m,\Sigma)$, measured by $\mathcal{G}$. This measure of non-Gaussianity is often denoted as the *negentropy*, e.g. [36]. (ii) From that nearest Gaussian to the corresponding uncorrelated Gaussian $\mathcal{N}(m,\mathrm{Diag}(\Sigma))$, measured by $\mathcal{C}$. (iii) From the uncorrelated Gaussian to standard normal.

We do not make explicit use of the fact that the *non-Standardness* can again be decomposed, but we show it nevertheless to relate our result to the literature: The Pythagorean identity $\mathcal{D}_{KL}(p(x)\|\mathcal{N}(m,\mathrm{Diag}(\Sigma))) = \mathcal{G}(p) + \mathcal{C}(p)$ has been shown before by [25, Section 2.3]. Both their and our result are specific applications of the general [37, Theorem 3.8] from information geometry. Our proof is given in Appendix B.1.

Proposition 1 is visualized in Figure 1. In an experiment, we fit a set of Glow [6] coupling flows of increasing depths to the EMNIST digit dataset [38] using maximum likelihood loss and measure the capability of each flow in decreasing $\mathcal{G}$ and $\mathcal{S}$ (Details in Appendix A.1). The form of the non-Standardness $\mathcal{S}$ is given by the well-known KL divergence between the involved normal distributions, see Equation (30) in Appendix B.1. It is invariant under rotations $Q$ and only depends on the first two moments $m, \Sigma$:

$$\mathcal{S}(m,\Sigma) := \mathcal{S}(p) = \frac{1}{2}(\|m\|^2 + \mathrm{tr}\,\Sigma - D - \log\det\Sigma) = \mathcal{S}(Qm, Q\Sigma Q^{\mathrm{T}}). \tag{8}$$

The non-Standardness $\mathcal{S}$ will be our measure on how far the covariance and mean have approached the standard normal in the latent space. We give explicit loss guarantees for $\mathcal{S}$ for a single coupling block in Theorems 1 and 2 and imply a linear convergence rate for a deep network in Theorem 3.

Deep Normalizing Flows are typically trained end-to-end, i.e. the entire stack of blocks is trained jointly. In this work, our ansatz is to consider the effect of a single coupling block on the non-Standardness $\mathcal{S}$. Then, we combine the effect of many isolated blocks, disregarding potential further improvements to $\mathcal{S}$ due to joint, cooperative learning of all blocks. This simplifies the theoretical analysis of the network, but it is not a restriction on the model: Any function that is achieved in block-wise training could also be the solution of end-to-end training.

We aim to strongly reduce $\mathcal{S}$ while leaving room for a complementary theory explaining how non-Gaussianity $\mathcal{G}$ is reduced in practice. Note that affine-linear functions $Ax + b$ can never change $\mathcal{G}$, because they jointly transform the distribution $p(x)$ at hand and correspondingly the closest Gaussian to it (see Lemma 1 in Appendix B.2). Thus, if we restrict our coupling layers to be affine-linear functions, we are able to reduce $\mathcal{S}$ without increasing $\mathcal{G}$ in turn. This motivates considering affine-linear couplings of the following form, spelled out together with ActNorm as given by Equation (5). **The results in this work apply to all coupling architectures**, as they all can represent this coupling, see Appendix C:

$$\begin{pmatrix} p_1 \\ a_1 \end{pmatrix} = (f_{\text{act}} \circ f_{\text{cpl}})(Qx) = r \odot \begin{pmatrix} I & 0 \\ T & I \end{pmatrix} \begin{pmatrix} p_0 \\ a_0 \end{pmatrix} + u. \tag{9}$$

For future work considering $\mathcal{G}$, we propose to lift the restriction to affine-linear layers while making sure that $\mathcal{S}$ behaves as described in what follows. As the convergence of $\mathcal{G}$ however will strongly depend on the coupling architecture and data $p(x)$ at hand, this is beyond the scope of this work.

Our first result shows which mean $m_1$ and covariance $\Sigma_1$ a single affine-linear coupling as in Equation (9) yields to minimize $\mathcal{S}(m_1, \Sigma_1)$ given data with mean $m$ and covariance $\Sigma$, rotated by $Q$:

**Proposition 2** (Proof in Appendix B.2). *Given $D$-dimensional data with mean $m$ and covariance $\Sigma$ and a rotation matrix $Q$. Split the covariance of the rotated data into four blocks, corresponding to the passive and active dimensions of the coupling layer:*

$$Q\Sigma Q^{\text{T}} = \Sigma_0 = \begin{pmatrix} \Sigma_{0,pp} & \Sigma_{0,pa} \\ \Sigma_{0,ap} & \Sigma_{0,aa} \end{pmatrix} \tag{10}$$

*Then, the moments $m_1, \Sigma_1$ that can be reached by a coupling as in Equation (9) are:*

$$m_1 = 0, \qquad \Sigma_1 = \begin{pmatrix} M(\Sigma_{0,pp}) & 0 \\ 0 & M(\Sigma_{0,aa} - \Sigma_{0,ap}\Sigma_{0,pp}^{-1}\Sigma_{0,pa}) \end{pmatrix}. \tag{11}$$

*This minimizes $\mathcal{S}$ as given in Equation (8), and $\mathcal{G}$ does not increase.*

The function $M$ takes a matrix $A$ and rescales the diagonal to 1 as follows. It is a well-known operation in numerics called Diagonal scaling or Jacobi preconditioning so that $M(A)_{ii} = 1$:

$$M(A)_{ij} = \sqrt{A_{ii}A_{jj}}^{-1} A_{ij} = (\text{Diag}(A)^{-1/2} A \, \text{Diag}(A)^{-1/2})_{ij}. \tag{12}$$

Proposition 2 shows how the covariance can be brought closer to the identity.

The new covariance has passive and active dimensions uncorrelated. In the active subspace, the covariance is the Schur complement $\Sigma_{0,aa} - \Sigma_{0,ap}\Sigma_{0,pp}^{-1}\Sigma_{0,pa}$. This coincides with the covariance of the Gaussian $\mathcal{N}(0, \Sigma)$ as it is conditioned on any passive value $p$. Afterwards, the diagonal is rescaled to one, matching the standard deviations of all dimensions with the desired latent code. The proof is based on a more general result how a single layer maximally reduces the Maximum Likelihood Loss for arbitrary data [14], which we apply to the non-Standardness $\mathcal{S}$ (see Appendix B.2).

Figure 2 shows an experiment in which a single affine-linear layer was trained to bring the covariance of EMNIST digits [38] as close to $I$ as possible (Details in Appendix A.2). The experimental result coincides with the prediction by Proposition 2. Due to the finite batch-size, a small difference between theory and experiment remains.

## 5 Explicit convergence rate

In Section 4, we showed how a single coupling layer acts on the first two moments of a given data distribution to whiten it. We now explicitly demonstrate how much progress this means in terms

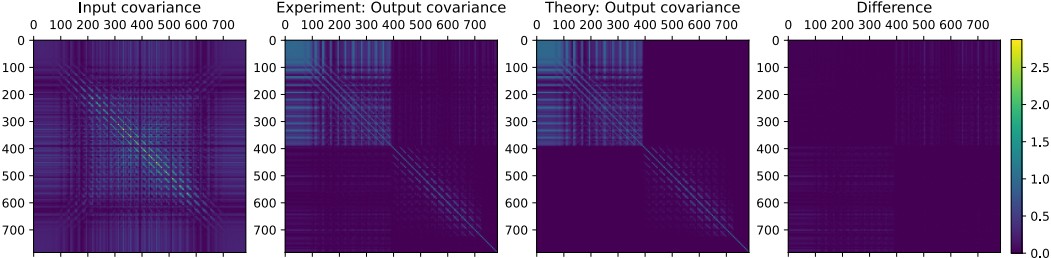

Figure 2: **How a single coupling layer can whiten the covariance** at the example of the EMNIST digits covariance matrix *(first panel)*. The covariance after a single layer trained experimentally to minimize non-Standardness $\mathcal{S}(m_1, \Sigma_1)$ *(second panel)*, which matches closely the prediction of Proposition 2 *(third panel)*. The difference between theory and experiment vanishes *(last panel)*.

of the non-Standardness $\mathcal{S}(m_1, \Sigma_1)$ averaged over rotations $Q$ (Theorems 1 and 2) and show the consequences for multiple blocks (Theorem 3).

## 5.1 Single coupling block guarantees

Proposition 2 allows the computation of the minimum non-Standardness after a single coupling block given its rotation $Q$, by evaluating $\mathcal{S}(m_1, \Sigma_1)$. In fact, if we were to choose $Q$ such that the data is rotated so that principal components lie on the axes (i.e. obtain $Q$ using PCA), a single coupling block suffices to reduce the covariance to the identity: $\Sigma_0 = Q\Sigma Q^T$ would be a diagonal matrix and $\Sigma_1 = I$. This is not the case in practice, where this optimal orientation has zero probability: $Q$ is chosen uniformly at random before training from all orthogonal matrices. One could argue that one should whiten the data before passing it to the flow, reducing $\mathcal{S}$ to zero from the start. However, any change in the architecture could possibly alter the performance of the network with regard to reducing the non-Gaussianity $\mathcal{G}$. Also, our work shows that coupling-based normalizing flows are already well-equipped to bring the non-Standardness to zero without such modifications. To properly describe the achievable non-Standardness $\mathcal{S}$, **we formulate all guarantees as expectations over the rotation** $Q$, corresponding to the loss averaged over training runs.

We make two mild assumptions on our data that are part of usual data-preprocessing, when the mean is subtracted from the data and all data points are divided by the scalar $\sqrt{\operatorname{tr} \Sigma / D}$ (not to be confused with diagonal preconditioning, which acts dimension-wise).

**Assumption 1.** *The data $p(x)$ is centered:* $\mathbb{E}_{x \sim p(x)}[x] = 0$.

**Assumption 2.** *The covariance is normalized:* $\operatorname{tr} \Sigma = D$.

The assumptions simplify the non-Standardness in Equation (8), which now only depends on the determinant of $\Sigma$:

$$\mathcal{S}(\Sigma) = -\tfrac{1}{2} \log \det \Sigma = -\tfrac{1}{2} \log \det \Sigma_0 = \mathcal{S}(\Sigma_0) \tag{13}$$

for arbitrary rotation $Q$. We aim to compute the average non-Standardness after a single block $\mathbb{E}_{Q \in p(Q)}[\mathcal{S}(\Sigma_1(Q))]$. For any $Q$, $\mathcal{S}(\Sigma_1)$ is again given by the determinant of the covariance $\Sigma_1(Q)$ as Assumptions 1 and 2 remain fulfilled: By Proposition 2 $m_1 = 0$ and the diagonal preconditioning $M$ ensures that the trace of $\Sigma_1$ is $D$. We write $\det(\Sigma_1)$ via $M_a$ and $M_p$, the diagonal matrices that make up the diagonal preconditioning in Equation (12), and use the Schur determinantal formula for the determinant of block matrices: $\det(\Sigma_{0,pp}) \det(\Sigma_{0,aa} - \Sigma_{0,ap}\Sigma_{0,pp}^{-1}\Sigma_{0,pa}) = \det(\Sigma_0) = \det(\Sigma)$ [39]. We thus get $\det(\Sigma_1) = \det(M_p \Sigma_{0,pp} M_p) \det(M_a(\Sigma_{0,aa} - \Sigma_{0,ap}\Sigma_{0,pp}^{-1}\Sigma_{0,pa})M_a) = \det(M_p^2) \det(M_a^2) \det(\Sigma)$. Inserting this into Equation (13), we find:

$$\mathcal{S}(\Sigma_1) = -\tfrac{1}{2}(\log \det \Sigma + \log \det M_p^2 + \log \det M_a^2) \leq \mathcal{S}(\Sigma_0) = \mathcal{S}(\Sigma). \tag{14}$$

The inequality $\mathcal{S}(\Sigma_1) \leq \mathcal{S}(\Sigma_0)$ holds because $\Sigma_1 = \Sigma_0$ is an admissible solution of the coupling layer optimization, but $\Sigma_1$ as given by Proposition 2 is a minimizer of $\mathcal{S}(\Sigma_1)$.

We average this quantity over training runs, i.e. over rotations $Q$:

$$\mathbb{E}_{Q \sim p(Q)}[\mathcal{S}(\Sigma_1)] = -\tfrac{1}{2}\big(\log \det \Sigma + \mathbb{E}_{Q \sim p(Q)}[\log \det M_p^2] + \mathbb{E}_{Q \sim p(Q)}[\log \det M_a^2]\big). \tag{15}$$

The main difficulty lies in the computation of $\mathbb{E}_{Q \sim p(Q)}[\log \det M_a^2]$. Here, we contribute the two strong statements Theorems 1 and 2 below.

### 5.1.1 Precise guarantee

The first result relies on projected orbital measures as developed by [40]. This theory describes the eigenvalues of submatrices of matrices in a random basis. We require such a result for integrating over $p(Q)$ in $\mathbb{E}_{Q \sim p(Q)}[\log \det M_a^2]$. In contrast to typical choices of $p(Q)$, the theory to this date only covers data rotated by unitary matrices.[1] To comply with [40], we make two more assumptions:

**Assumption 3.** *The distribution of rotations is the Haar measure over* unitary *matrices* $U(D)$.

**Assumption 4.** *The eigenvalues of the covariance matrix* $\Sigma$ *are distinct:* $\lambda_i \neq \lambda_j$ *for* $i \neq j$.

One could think that the step from orthogonal to unitary rotations takes us far away from the scenario we want to consider. We will later observe empirically that the difference between averaging over unitary and orthogonal matrices is negligible. Technically, the covariance matrix remains positive definite, so the non-Standardness $\mathcal{S}$ is always real (see Appendix B.3.4). We will write $\mathbb{E}_{Q \sim U(D)}[\,\cdot\,]$ to denote expectations over unitary matrices.

Assumption 4 is typically satisfied when working with real data that are in 'general position'. We are now ready to **compute the average training performance** of a single coupling block:

**Theorem 1** (Proof in Appendix B.3). *Given $D$-dimensional data with covariance $\Sigma$ with eigenvalues $\lambda_1, \ldots \lambda_D$. Assume that Assumptions 1 to 4 hold. Then, after a single coupling block, the expected non-Standardness is bounded from above:*

$$\mathbb{E}_{Q \in U(D)}[\mathcal{S}(\Sigma_1(Q))] < \mathcal{S}(\Sigma) + \frac{D}{2}\log\left((-1)^{\frac{D}{2}+1}\sum_{i=1}^{D}\lambda_i^{1-\frac{D}{2}}\log(\lambda_i)R(\lambda_i^{-1};\lambda_{\neq i}^{-1})e_{\frac{D}{2}-1}(\lambda_{\neq i}^{-1})\right). \quad (16)$$

*Here, $\lambda_{\neq i} := \{\lambda_1, \ldots, \lambda_{i-1}, \lambda_{i+1}, \ldots, \lambda_D\}$ and $R, e_K$ are given by:*

$$R(a; \{b_i\}_{i=1}^{N}) = \prod_{i=1}^{N}\frac{1}{a-b_i} \quad and \quad e_K(\{b_i\}_{i=1}^{N}) = \sum_{0<i_1<\cdots<i_K\leq N} b_{i_1}\cdots b_{i_K}. \quad (17)$$

Inequality (16) sharply bounds the expected non-Standardness that can be achieved by a single block. The only approximation made is an inequality which comes close to equality as the dimension $D$ increases due to the concentration of the corresponding probability distribution.

Figure 3 shows an experiment confirming Theorem 1 (Details in Appendix A.3). We start with covariance matrices using parametrized eigenvalue spectra. On each, we first apply a single coupling block with random Q and train the coupling that maximally reduces $\mathcal{S}$ (Proposition 2). Then we iteratively append 32 additional blocks in the same manner, building a flow of that depth. We average the resulting empirical ratio $\mathcal{S}(\Sigma_1)/\mathcal{S}(\Sigma)$ over several *orthogonal* orientations $Q$ of the rotation layer for each input covariance matrix. Then, we compare this to (i) experimentally averaging over *unitary* rotations and (ii) to the prediction by Theorem 1 and confirm that it is a valid and close upper bound. Details for replication and more examples can be found in Appendix A.3.

The proof explicitly integrates $\mathbb{E}[M_a^2]$ using [40] (see Appendix B.3). Numerically evaluating Equation (16) can be hard even for small $D$ as the summands scale as $\mathcal{O}(\exp(D))$, but the overall sum scales as $\mathcal{O}(D)$. High values cancel due to $R$ alternating in sign, and one requires arbitrary-precision floating point software to evaluate Equation (16).

### 5.1.2 Interpretable guarantee

The guarantee in Theorem 1 yields useful predictions, but it does not lend itself to further analysis: How does the bound behave over several coupling blocks? What is the behavior for varying dimension $D$? Also, Assumption 3 restricts formal reasoning as we are interested in averaging over orthogonal and not unitary rotations. Our second single-block guarantee depends only on simple metrics of the covariance. Moreover, we drop Assumptions 3 and 4, averaging over orthogonal, not unitary, $Q$:

---

[1]The only result known to us would yield predictions for $D = 2$ [41], whereas we are interested in large $D$.

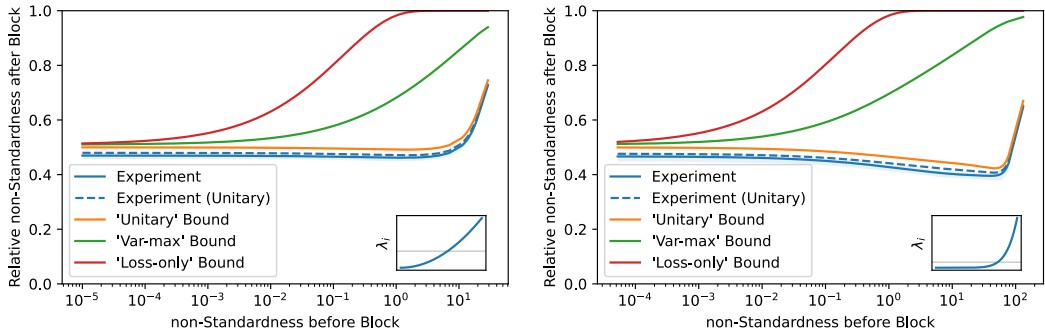

Figure 3: **Comparison between predicted non-Standardness and experiment** for 48-dimensional parametrized eigenvalue spectra *(insets)*, varied over a parameter which controls the spread of the spectrum and thus changes $\mathcal{S}$. The experimental average over *orthogonal* rotations matrices (blue, shaded by Interquartile Range IQR) is closely matched by the experimental average over *unitary* matrices (dotted blue). The prediction by Theorem 1 is a close upper bound that closely matches the experimental behavior (orange). The predictions by Theorem 2 are less precise, but converge to the same value as the precise bound for covariances close to the identitiy: 'Var-max' is Equation (18a) (green) and 'Loss-only' is Equation (18b) (red). More details and examples in Appendix A.3.

**Theorem 2** (Proof in Appendix B.4). *Given $D$-dimensional data fulfilling Assumptions 1 and 2 with covariance $\Sigma \neq I$ with eigenvalues $\lambda_1, \ldots \lambda_D$. Then, after a single coupling block, the expected loss can be bounded from above:*

$$\mathbb{E}_{Q \in O(D)}[\mathcal{S}(\Sigma_1(Q))] \leq \mathcal{S}(\Sigma) + \frac{D}{4} \log\left(1 - \frac{D^2}{2(D-1)(D+2)} \frac{\mathrm{Var}[\lambda]}{\lambda_{\max}}\right) \tag{18a}$$

$$\leq \mathcal{S}(\Sigma) + \frac{D}{4} \log\left(1 - \frac{D^2}{(D-1)(D+2)} \frac{1 - \sqrt{1 - g^D}}{1 + \sqrt{1 - g^D}}(1 - g)\right) < \mathcal{S}(\Sigma). \tag{18b}$$

*Here, $g$ is the geometric mean of the eigenvalues: $g = \prod_{i=1}^{D} \lambda_i^{1/D} = \exp(-2\mathcal{S}(\Sigma)/D) < 1$ which is a bijection of $\mathcal{S}(\Sigma)$.*

These two new bounds on the average achievable non-Standardness $\mathcal{S}$ after a single block are also depicted in Figure 3. They make useful predictions, but are less precise than Theorem 1. The second bound will be especially useful in what follows because it only depends on the non-Standardness before the block $\mathcal{S}(\Sigma)$.

The full proof is given in Appendix B.4. It relies on the integration of monomials of entries of random orthogonal matrices as described by [42] and the arithmetic mean-geometric mean inequality by [43].

The first bound suggests an important property of the non-Standardness convergence of a coupling-based normalizing flow in terms of dimension: The performance only marginally depends on the dimension. To see this, divide Equation (18a) by $D$ to obtain a statement about the non-Standardness per dimension $\mathcal{S}/D$. Then take several data sets with different dimension but same spectrum characteristics (i.e. same geometric mean, variance and maximum of covariance eigenvalues). The guarantee is then approximately constant in $D$ (it varies slightly with $D^2/(D^2 + D - 2)$, which is always close to 1).

## 5.2 Deep network guarantee

The previous Section 5.1 was concerned with determining how much a *single* coupling block can typically contribute towards reducing the $\mathcal{S}$ to zero. Now, we extend this result to compute the expected non-Standardness after a *deep* coupling-based normalizing flow as an explicit function of the number of blocks. We again treat the rotation layer of each block as a random variable, as it is randomly determined before training.

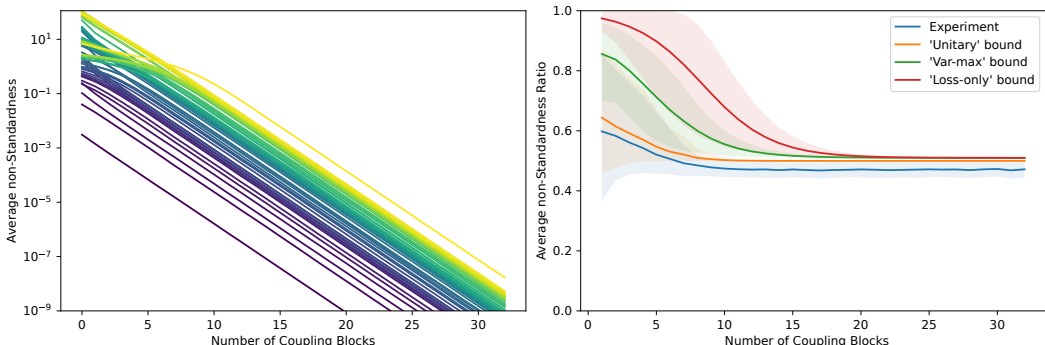

Figure 4: **Deep network convergence of covariance on toy dataset.** *(Left)* Each line shows the experimental convergence of $\mathcal{S}$ via the repeated application of Proposition 2, averaged over 32 runs with different rotations $Q$. *(Right)* The empirical convergence rate (blue), i.e. the ratio of $\mathcal{S}$ before and after a block, is correctly bounded from above by our predictions in Theorem 1 (orange), and the bounds in Theorem 2: Equation (18a) (green) and Equation (18b) (red). The solid lines show the ratio (bounds) averaged over the toy dataset and rotations, the shade is the IQR. The experiment suggests that a convergence rate like Theorem 3 can also be derived for the remaining bounds.

We find that the **convergence rate** of the covariance to the identity is (at least) **linear**:

**Theorem 3** (Proof in Appendix B.5). *Given $D$-dimensional data fulfilling Assumptions 1 and 2 with covariance $\Sigma$. Then, after $L$ coupling blocks, the expected loss is smaller than:*

$$\mathbb{E}_{Q_1,\ldots,Q_L \in O(D)}[\mathcal{S}(\Sigma_L)] \leq \gamma\big(\mathcal{S}(\Sigma)\big)^L \mathcal{S}(\Sigma), \tag{19}$$

*where the convergence rate depends on the non-Standardness before training:*

$$\gamma(\mathcal{S}) = 1 + \frac{1}{4\mathcal{S}/D} \log\left(1 - \frac{D^2}{(D-1)(D+2)} \frac{1 - \sqrt{1 - g(\mathcal{S})^D}}{1 + \sqrt{1 - g(\mathcal{S})^D}} \big(1 - g(\mathcal{S})\big)\right) < 1. \tag{20}$$

The non-Standardness decreases at least exponentially fast in the number of blocks. The convergence rate that holds for a deep network is computed using the non-Standardness of the input data $\mathcal{S}(\Sigma)$. This rate comes from Equation (18b). The proof uses that $\gamma(\mathcal{S})$ improves from block to block as $\mathcal{S}$ decreases (see Appendix B.5). Again, $g(\mathcal{S}) = \exp(-2\mathcal{S}/D) < 1$ is the geometric mean of eigenvalues of $\Sigma$, which increases from block to block.

Figure 4 shows the convergence of the non-Standardness to zero in an experiment. We build a toy dataset of various covariances where we aim to capture a plethora of possible cases (see Appendix A.4). We apply a single coupling block with random $Q$ and the coupling that maximally reduces $\mathcal{S}$ via Proposition 2. We iteratively add such blocks 32 times, building a flow of that depth. The resulting convergence of $\mathcal{S}$ as a function of depth is averaged over 32 runs with different rotations. The measured curve confirms Theorem 3. We find that the rate $\gamma$ in Equation (20) is correct, but several experiments show even faster convergence in practice. Indeed, the experiments suggest that dividing all upper bounds for $\mathbb{E}[\mathcal{S}(\Sigma_1)]$ in Theorems 1 and 2 by $\mathcal{S}(\Sigma)$ also bounds the non-Standardness ratio for subsequent blocks. Formally, we conjecture that $\mathbb{E}[\mathcal{S}(\Sigma_L)]/\mathcal{S}(\Sigma) \leq (B/\mathcal{S}(\Sigma))^L$ where $B$ is the rhs. of Equations (16) and (18a) (Theorem 3 shows exactly this for Equation (18b)). We leave a proof or falsification of this conjecture open to future work.

The experiment also suggests that all bounds agree after a few blocks, leaving a small gap to the experiment. We can explicitly compute this limit value of $\gamma(\mathcal{S})$ by taking $\mathcal{S} \to 0$:

$$\gamma(\mathcal{S}) \xrightarrow{\mathcal{S} \to 0} \frac{D(D+2)-4}{2(D-1)(D+2)} \in \big[1/2, 5/9\big]. \tag{21}$$

The two experimental observations together with this limit value suggest the heuristic that **a single additional coupling block typically reduces the non-Standardness $\mathcal{S}$ by a factor of approximately 50%** if previous blocks are left unchanged, and possibly faster if cooperations between blocks are taken into account.

# 6   Conclusion

To the best of our knowledge, this is the first work on coupling-based normalizing flows that provides a quantitative convergence analysis in terms of the KL divergence. Specifically, a minimal convergence rate is established at which flows whiten the covariance of the input data under this strong measure of discrepancy of probability distributions. Splitting the loss into the non-Gaussianity (negentropy) $\mathcal{G}$ and the non-Standardness $\mathcal{S}$, we show that this whitening is a necessary condition for the flow to converge and give explicit guarantees. Our derivations suggest the rule of thumb that $\mathcal{S}$ can typically be reduced by about 50% per coupling block.

Our central idea was to separate out the contribution a single isolated block can make to reduce the loss, arguing that end-to-end training can only outperform the concatenation of isolated blocks.

Having separated the tasks a normalizing flow has to solve, and having explained how the non-Standardness $\mathcal{S}$ can be reduced to zero, we hope that explaining also the entire convergence of $\mathcal{L} = \mathcal{G} + \mathcal{S}$ with respect to the KL divergence is within reach. In particular, our theory did not yet explore how the non-linear part of each coupling block reduces the non-Gaussianity $\mathcal{G}$.

## Acknowledgments and Disclosure of Funding

This work is supported by Deutsche Forschungsgemeinschaft (DFG, German Research Foundation) under Germany's Excellence Strategy EXC-2181/1 - 390900948 (the Heidelberg STRUCTURES Cluster of Excellence). It is also supported by the Vector Stiftung in the project TRINN (P2019-0092).

We thank our colleagues (in alphabetical order) Marcel Meyer, Jens Müller, Robert Schmier and Peter Sorrenson for their help and fruitful discussions.

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
