# Whitening Convergence Rate of Coupling-based Normalizing Flows: Appendix

## A  Details on Experiments

All experiments were carried out on a single AMD Ryzen 7 3700X 8-Core Processor together with a NVIDIA GeForce RTX 2080. At `https://github.com/vislearn/Coupling-Flow-Bound` we have made available the code for all experiments.

### A.1  Deep network on EMNIST

In this experiment, we estimate the capability of affine normalizing flows in reducing the non-Standardness $\mathcal{S}$ (see Equation (8)) as a function of the number of layers. We compare this to the theoretic bound in Theorem 1.

To this end, we train affine normalizing flows on EMNIST digits [38]. We leverage a 20-block Glow architecture as described in Section 3. To measure the effect of depth $L = 1, \ldots, 20$ of the flow on $\mathcal{S}$, we truncate the architecture to $L$ layers.

The architecture is built as follows: We start by down-sampling the input image from gray scale $1 \times 28 \times 28$ to $4 \times 14 \times 14$: Each group of four neighboring pixels is reordered into one pixel with four times the channels in a checkerboard-like pattern. Then, eight convolutional coupling blocks with 16 hidden channels are applied. They are followed by another down-sampling to $16 \times 7 \times 7$ and eight convolutional coupling blocks with 32 hidden channels. After flattening the input, four fully-connected affine coupling blocks are added with 392 hidden dimensions.

When truncating this architecture, we remove blocks *from the left*. For example, when one block is present ($L = 1$), only the last coupling block with the fully connected subnetwork remains. This makes the theory in this paper applicable, as Proposition 2 assumes that the neural networks $s$ and $t$ are fully connected (otherwise, the whitening operation cannot always be represented).

We train each depth from scratch for 300 epochs using Adam with a learning rate of $3 \cdot 10^{-3}$ which is reduced by a factor of .1 after 100 and 200 epochs. The batch size is 240 which implies 1000 iterations per epoch.

Given the 20 networks of different depth, we split the loss into the non-Gaussianity $\mathcal{G}$ and non-Standardness $\mathcal{S}$ as suggested by Proposition 1. To do so, we compute the empirical covariances $\Sigma_l$ of 10'000 test samples pushed through each flow.

To relate this experiment to our theory, we take the covariance matrices obtained using the trained flows $\Sigma_l$ and apply Theorem 1 on each. This yields an upper bound on the expected non-Standardness after training another network with depth increased by one. In other words, given $\Sigma_l$, Theorem 1 predicts an upper bound on the expected $\mathbb{E}_{Q_{l+1} \sim p(Q)}[\mathcal{S}(\Sigma_{l+1})]$. We observe that the experimentally observed non-Standardness behaves similar to the upper bound. We do not expect this to be the case in general: There might be a trade-off between reducing $\mathcal{S}$ and $\mathcal{G}$, so the optimization might actually decide for reducing $\mathcal{G}$ at the cost of increasing $\mathcal{S}$. We only show that with the covariance in Proposition 2, $\mathcal{G}$ does not increase. On the other end, an affine flow might actually be able to reduce the non-Standardness stronger than predicted, as our theory does not take potentially useful cooperation between layers into account.

We average all results over eight runs per depth (i.e. $8 \cdot 20 = 160$ networks in total). Despite different random orientations in each run, the results are very concentrated: We find error bars so small that they are not visible in Figure 1.

We observe that after four blocks, the non-Standardness is close to zero. Here, the flow consists of four coupling blocks with fully connected subnetworks. This justifies the use of convolutional networks for $s$ and $t$ in the remaining blocks; they can only reduce correlations between pixels locally, thus not reducing non-Standardness as strongly as predicted. However, the non-Standardness is reduced enough by only four coupling blocks.

Figure 5 shows samples from one networks trained for each depth (sampling temperature $T = 0.7$).

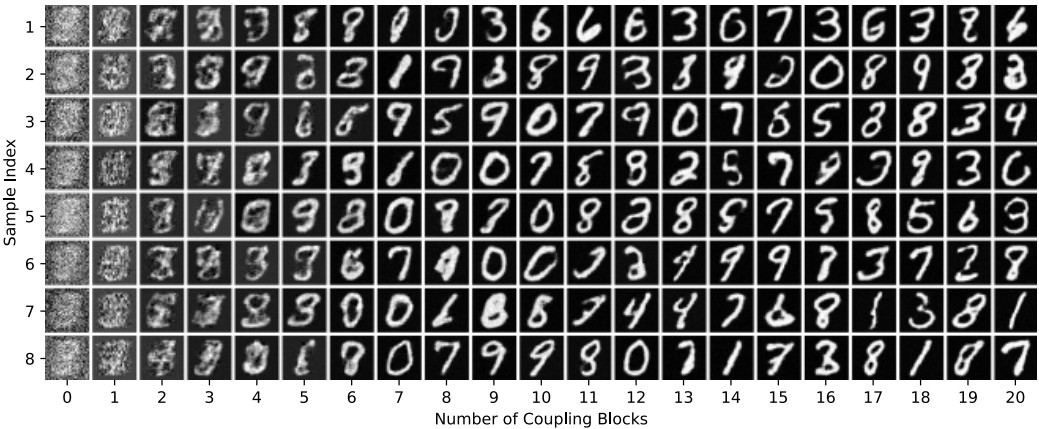

Figure 5: Samples generated by the affine coupling flows with varying depth trained for Figure 1. Each column shows eight samples by a network of the corresponding depth.

## A.2 Single layer on EMNIST digit covariance

This experiment confirms that the covariance minimizing the non-Standardness $\mathcal{S}(\Sigma_1)$ after a single layer is correctly predicted by Proposition 2.

To get an interesting covariance matrix, we flatten the EMNIST digits training data and compute its covariance matrix $\Sigma$, as depicted in Figure 2 on the left. We then sample a multivariate Gaussian with this covariance matrix and train a single affine coupling layer. As the data is Gaussian, we can train with the standard maximum likelihood loss as it is equivalent to the non-Standardness $\mathcal{S}$. We use Adam with a learning rate $0.05$, a batch size of $2048$ and train for $512$ iterations.

## A.3 Single block on toy data

This experiment explores the average non-Standardness that can be reached by a single layer by modifying the covariance as given by Proposition 2. It also aims to confirm the upper bounds shown in Theorems 1 and 2.

We build a family of toy covariance matrices to work with. As the data will be randomly rotated anyway, we choose the matrices to be diagonal w.l.o.g., i.e. we directly design the eigenvalue spectrum of each covariance. We prescribe this spectrum by a continuous function $\mu : [0,1] \to \mathbb{R}_+$. It is chosen bijective to ensure that the eigenvalues are distinct. We then define the eigenvalues as follows:

$$\mu_i = \mu\left(\tfrac{i}{D-1}\right) \quad i = 0, \ldots, D-1. \tag{22}$$

With this approach, we can systematically modify eigenvalue/noise spectra.

Given a vector of eigenvalues $(\mu_i)_i$, we need to ensure that its mean is one. We do so by dividing by the mean:

$$\nu_i := \frac{\mu_i}{\sum_{i=1}^{D} \mu_i / D}. \tag{23}$$

Finally, we add a scaling parameter $s > 0$ that defines how close the spectrum is to the identity:

$$\lambda_i^{(s)} := (\nu_i - 1) \cdot s + 1. \tag{24}$$

The non-Standardness strictly decreases as $s$ comes closer to $0$. As the eigenvalues always have to be positive, $s$ must be chosen smaller than:

$$s < \frac{1}{1 - \lambda_{\min}} =: s_{\max}. \tag{25}$$

Given a spectrum $\lambda_i^{(s)}$, we build a diagonal covariance matrix

$$\Sigma = \mathrm{Diag}(\lambda_i^{(s)})_{i=1}^{D}. \tag{26}$$

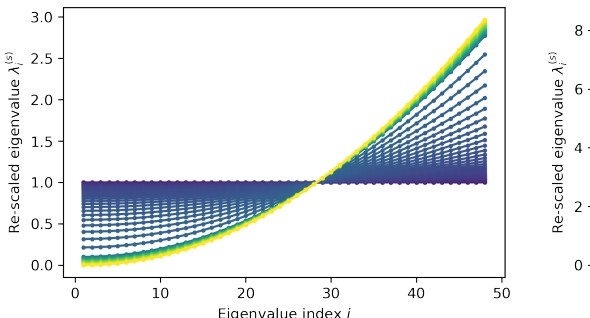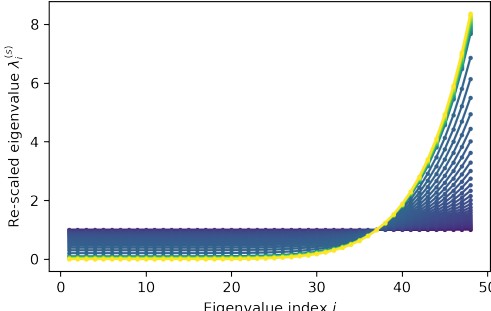

Figure 6: Eigenvalue spectra used for experiment depicted in Figure 3. *(Left)* $\mu(x) = x^2$ and *(right)* $\mu(x) = x^8$. Each line corresponds to a different scaling $s$.

For the experimental baseline, we sample $N_{\text{rot}}$ orthogonal and unitary rotation matrices $Q \sim p(Q)$ from the corresponding Haar measure over $O(D)$ and $U(D)$. We employ `scipy.stats.ortho_group` respectively `scipy.stats.unitary_group`. This yields the covariance of the rotated data:

$$\Sigma_0 = Q\Sigma Q^T. \tag{27}$$

(Or, $Q^*$ instead of $Q^T$ if we average over unitary matrices).

We do not train affine coupling layers directly. Instead, we make use of the single layer output covariance $\Sigma_1$ from Proposition 2.

We choose the following numerical values for $s$: To get a close look at the case where $s \to 0$ and correspondingly $\mathcal{S} \to 0$, we take $N_{\text{scale}}/3$ geometrically spaced points in $[0.001 s_{\max}, 0.9 s_{\max}]$. To accurately capture the off-minimum behavior, we add to that $2N_{\text{scale}}/3$ linearly spaced points between $[0.9 s_{\max}, .999 s_{\max}]$.

We choose $N_{\text{rot}} = 100$ and $N_{\text{scale}} = 150$ for all experiments. To save computational resources, we re-use the rotations sampled for the first scale for the remaining.

In Figure 3, we showed the experiment for the parameterized spectra $\mu(x) = x^2$ and $\mu(x) = x^8$. For both, Figure 6 shows which rescaled eigenvalue spectra were used in this experiment. In Figure 7, we give examples for more spectra.

### A.4 Layer-wise training on toy data

In this experiment, we track the non-Standardness as layers are added, check Theorem 3, and compare the convergence rate Equation (20) to the other bounds in Theorems 1 and 2.

This experiment uses a different set of toy covariances than Appendix A.3. This time, we build a plethora of different initial covariances (eigenvalue spectra) that include extreme cases:

1. All eigenvalues are set to 1 except for one that is varying.

2. All eigenvalues have the same value that is varied, except for one that is set to 1.

3. Split the eigenvalues into two halves, respectively having the same value: The first half is varied, the second half assume the inverse value of the first half.

4. Randomly sample all eigenvalues uniformly from $[0, 2]$.

5. Randomly sample all eigenvalues between such that the logarithm is uniformly distributed over $[1/v_{\max}, v_{\max}]$.

Whenever we vary the value of any eigenvalue, we take $N_{\text{vary}}$ scalars geometrically spaced between $1/v_{\max}$ and $v_{\max}$. We exclude the case where all eigenvalues are equal to 1, implying a non-Standardness of 0.

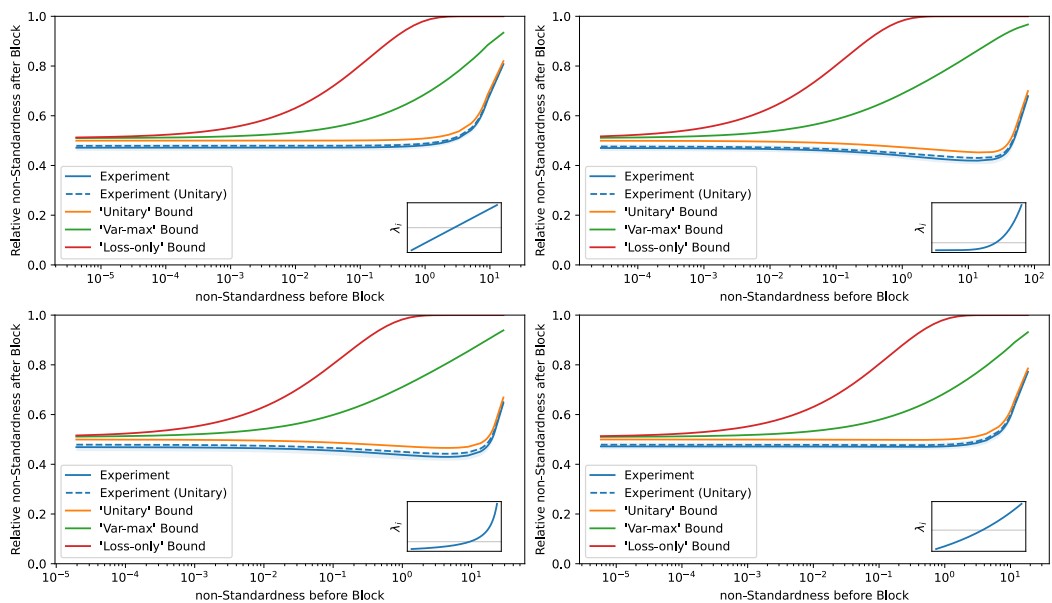

Figure 7: Examples for single layer relative non-Standardness on more eigenvalue spectra: *(Top left)* $\mu(x) = x$, *(top right)* $\mu(x) = x^5$, *(bottom left)* $\mu(x) = \frac{1}{1.1-x}$, *(bottom right)* $\mu(x) = \exp(x)$. More details in Figure 3.

To fulfill Assumption 4, we do not actually assign the same value to eigenvalues, but multiply them each with a linearly increasing factor in $(1 - \epsilon, 1 + \epsilon)$. We do not observe any change in experimental behavior from this, but this allows us evaluating Theorem 1.

Given the dataset of eigenvalues, we build diagonal covariances, repeatedly apply random rotations and the whitening procedure in Proposition 2. The details are given in Algorithm 1. For each input covariance, we obtain $N_{\mathrm{rot}}$ trajectories of covariances.

---

**Algorithm 1** Multi-layer non-Standardness experiment

---

**Input:** Input covariances $\Sigma^{(i)}, i = 1, \ldots, N$, number of rotations $N_{\mathrm{rot}}$, number of layers $L$.

   $\Sigma_0^{(i,r)} \leftarrow \Sigma^{(i)}$ **for** $i = 1, \ldots, N; r = 1, \ldots, N_{\mathrm{rot}}$ {Copy each input covariance $N_{\mathrm{rot}}$ times}
   **for** $l = 1, \ldots, L$ **do**
      $Q^{(r)} \sim O(D)$ **for** $r = 1, \ldots, N_{\mathrm{rot}}$ {Sample rotations}
      $\Sigma_{l-1}^{(i,r)\prime} \leftarrow Q^{(r)} \Sigma_{l-1}^{(i,r)} (Q^{(r)})^{\mathrm{T}}$ **for** $i = 1, \ldots, N; r = 1, \ldots, N_{\mathrm{rot}}$ {Apply rotations}
      $\Sigma_l^{(i,r)} \leftarrow$ Proposition 2 on $\Sigma_{l-1}^{(i,r)\prime}$ {Apply whitening step}
   **end for**
**Output:** $\{\Sigma_l^{(i,r)}\}_{l=1}^L$ **for** $i = 1, \ldots, N; r = 1, \ldots, N_{\mathrm{rot}}$.

---

We evaluate the non-Standardness of each covariance matrix $\mathcal{S}(\Sigma_l^{(i,r)})$ and average over rotations. This is shown in the left plot in Figure 4.

In addition, we compute the relative non-Standardness between layers:

$$\mathcal{S}(\Sigma_l^{(i,r)})/\mathcal{S}(\Sigma_{l-1}^{(i,r)}). \tag{28}$$

This quantity is averaged over rotations $r$ and instances $i$. It is depicted together with the corresponding interquartile range (IQR) in the right half of Figure 4.

We also evaluate each of the bounds on $\mathbb{E}_Q[\mathcal{S}(\Sigma_{l+1}^{(i,r)})]$ in Theorems 1 and 2 given $\Sigma_l^{(i,r)}$ and divide it by the non-Standardness $\mathcal{S}(\Sigma_l^{(i,r)})$. Again, we average over rotations and iterations.

Averaging over rotations might be counter-intuitive as the bounds explicitly calculate a value that is an average: It is necessary because for each initial covariance, we have $N_{\text{rot}}$ trajectories with different convergence behavior. Let us make this explicit. Denote by $B$ any of the bounds in Theorems 1 and 2:

$$\frac{\mathbb{E}_{Q_{l+1} \sim p(Q)}[\mathcal{S}(\Sigma_{l+1}^{(i,r)}(Q_{l+1}))]}{\mathcal{S}(\Sigma_l^{(i,r)})} \leq \frac{B(\Sigma_l^{(i,r)})}{\mathcal{S}(\Sigma_l^{(i,r)})}. \tag{29}$$

We average the quantity on the right over the different trajectories, i.e. over $i, r$. It only depends on the covariances in the $l$th layer in contrast to the expression on the left.

As hyperparameters to the experiment, we choose $D = 48, L = 32, N_{\text{vary}} = 128, N_{\text{rot}} = 32, v_{\text{max}} = 1000, \epsilon = 10^{-5}$. We stop a trajectory once the non-Standardness falls below $10^{-9}$ to avoid numerical instabilities.

# B  Detailed proofs

## B.1  Proof of Proposition 1

The explicit form of the non-Standardness is given by the KL divergence between the two multivariate Gaussians $\mathcal{N}(m, \Sigma)$ and $\mathcal{N}(0, I)$:

$$\mathcal{D}_{\text{KL}}(\mathcal{N}(m, \Sigma) \| \mathcal{N}(0, I)) \tag{30a}$$
$$= \mathbb{E}_{x \sim \mathcal{N}(m, \Sigma)}[\log \mathcal{N}(x; m, \Sigma) - \log \mathcal{N}(x; 0, I)] \tag{30b}$$
$$= \mathbb{E}_{x \sim \mathcal{N}(m, \Sigma)}[-\tfrac{1}{2} \log \det(2\pi\Sigma) - \tfrac{1}{2}(x - m)^{\text{T}}\Sigma^{-1}(x - m) + \tfrac{1}{2} \log \det(2\pi I_D) + \tfrac{1}{2}\|x\|^2] \tag{30c}$$
$$= \tfrac{1}{2}\left(-\log \det(\Sigma) + \mathbb{E}_{x \sim \mathcal{N}(m, \Sigma)}[-\tfrac{1}{2}(x - m)^{\text{T}}\Sigma^{-1}(x - m) + \tfrac{1}{2}\|x\|^2]\right) \tag{30d}$$
$$= \tfrac{1}{2}(\|m\|^2 + \operatorname{tr}\Sigma - D - \log \det\Sigma). \tag{30e}$$

*Proof.* We start with the first decomposition in Equation (6).

$$\mathcal{D}_{\text{KL}}(p \| \mathcal{N}(0, I)) - \mathcal{D}_{\text{KL}}(p \| \mathcal{N}(m, \Sigma)) \tag{31a}$$
$$= \mathbb{E}_{x \sim p(x)}[\log p(x) - \log \mathcal{N}(x; 0, I) - \log p(x) + \log \mathcal{N}(x; m, \Sigma)] \tag{31b}$$
$$= \mathbb{E}_{x \sim p(x)}[-\log \mathcal{N}(x; 0, I) + \log \mathcal{N}(x; m, \Sigma)] \tag{31c}$$
$$= \frac{1}{2}\mathbb{E}_{x \sim p(x)}[D \log(2\pi) + \|x\|^2 - D \log(2\pi) - \log \det\Sigma - (x - m)^{\text{T}}\Sigma^{-1}(x - m)] \tag{31d}$$
$$= \frac{1}{2}\mathbb{E}_{x \sim p(x)}[\|x\|^2 - \log \det\Sigma - (x - m)^{\text{T}}\Sigma^{-1}(x - m)] \tag{31e}$$
$$= \frac{1}{2}\left(\mathbb{E}_{x \sim p(x)}[\|x\|^2] - \log \det\Sigma - \mathbb{E}_{x \sim p(x)}[(x - m)^{\text{T}}\Sigma^{-1}(x - m)]\right). \tag{31f}$$

The open expectation values read:

$$\mathbb{E}_{x \sim p(x)}[\|x\|^2] = \mathbb{E}_{x \sim p(x)}[\sum_{i=1}^{D} x_i^2] = \sum_{i=1}^{D} \mathbb{E}_{x \sim p(x)}[x_i^2] = \sum_{i=1}^{D}(m_i^2 + \Sigma_{ii}) = \|m\|^2 + \operatorname{tr}\Sigma, \tag{32}$$

and interpreting $(x - m)$ as a $\mathbb{R}^{D \times 1}$ matrix, we can re-write using the trace. Then use the cyclic property and linearity of the trace:

$$\mathbb{E}_{x \sim p(x)}[(x - m)^{\text{T}}\Sigma^{-1}(x - m)] = \mathbb{E}_{x \sim p(x)}[\operatorname{tr}((x - m)^{\text{T}}\Sigma^{-1}(x - m))] \tag{33a}$$
$$= \mathbb{E}_{x \sim p(x)}[\operatorname{tr}((x - m)(x - m)^{\text{T}}\Sigma^{-1})] \tag{33b}$$
$$= \operatorname{tr}(\mathbb{E}_{x \sim p(x)}[(x - m)(x - m)^{\text{T}}]\Sigma^{-1}) \tag{33c}$$
$$= \operatorname{tr}(\Sigma\Sigma^{-1}) \tag{33d}$$
$$= D. \tag{33e}$$

Inserting the two expectation values, we identify:

$$\mathcal{D}_{\text{KL}}(p\|\mathcal{N}(0,I)) - \mathcal{D}_{\text{KL}}(p\|\mathcal{N}(m,\Sigma)) = \frac{1}{2}\Big(\|m\|^2 + \operatorname{tr}\Sigma - \log\det\Sigma - D\Big) \tag{34a}$$

$$= \mathcal{D}_{\text{KL}}(\mathcal{N}(m,\Sigma)\|\mathcal{N}(0,I)), \tag{34b}$$

and obtain Equation (6).

Now we move on to show Equation (7):

$$\mathcal{C}(p) = \mathcal{D}_{\text{KL}}(\mathcal{N}(m,\Sigma)\|\mathcal{N}(m,\operatorname{Diag}(\Sigma))) \tag{35a}$$

$$= \frac{1}{2}\left(\operatorname{tr}\Big((\operatorname{Diag}\Sigma)^{-1}\Sigma\Big) - D + \log\frac{\det(\operatorname{Diag}(\Sigma))}{\det\Sigma}\right) \tag{35b}$$

$$= \frac{1}{2}\log\frac{\det(\operatorname{Diag}(\Sigma))}{\det\Sigma}, \tag{35c}$$

and

$$\mathcal{D}_{\text{KL}}(\mathcal{N}(m,\operatorname{Diag}(\Sigma))\|\mathcal{N}(0,I)) = \frac{1}{2}\Big(\operatorname{tr}\operatorname{Diag}(\Sigma) - D - \log(\det(\operatorname{Diag}(\Sigma)))\Big) \tag{36a}$$

$$= \frac{1}{2}\Big(\operatorname{tr}\Sigma - D - \log(\det(\operatorname{Diag}(\Sigma)))\Big). \tag{36b}$$

Adding the two divergences yields Equation (7). $\square$

## B.2 Proof of Proposition 2

We first show that an affine-linear function $g(x)$ as assumed in Proposition 2 cannot change the non-Gaussianity $\mathcal{G}$. This has already been argued by [36]. We provide an explicit proof:

**Lemma 1.** *Given a $D$-dimensional distribution and an affine-linear function*

$$g(x) = Ax + b \tag{37}$$

*for some $A \in \mathbb{R}^{D\times D}$ with $\det A > 0$ and $b \in \mathbb{R}^D$. Then:*

$$\mathcal{G}(g_\sharp p) = \mathcal{G}(p). \tag{38}$$

*Proof.* The non-Gaussianity $\mathcal{G}$ is given by:

$$\mathcal{G}(p) = \mathcal{D}_{\text{KL}}(p(x)\|\mathcal{N}(m,\Sigma)). \tag{39}$$

Mean and covariance of the push-forward of $p$ via $g$ read:

$$\mathbb{E}_{x\sim p(x)}[g(x)] = \mathbb{E}_{x\sim p(x)}[Ax + b] = Am + b = m_1, \tag{40a}$$

$$\operatorname{Cov}_{x\sim p(x)}[g(x)] = \operatorname{Cov}_{x\sim p(x)}[Ax + b] = A\Sigma A^{\text{T}} = \Sigma_1. \tag{40b}$$

Thus, the non-Gaussianity after applying $g$ reads:

$$\mathcal{G}(g_\sharp p) = \mathcal{D}_{\text{KL}}(g_\sharp p\|\mathcal{N}(m_1,\Sigma_1)). \tag{41}$$

The push-forward of $\mathcal{N}(m,\Sigma)$ via $g$ is identical to the normal distribution that occurs in the non-Gaussianity of $g_\sharp p$:

$$g_\sharp \mathcal{N}(m,\Sigma) = \mathcal{N}(m_1,\Sigma_1), \tag{42}$$

Now, we make use of the fact that the KL divergence is invariant if both arguments are transformed by any invertible function $g$:

$$\mathcal{D}_{\text{KL}}(p_1(x)\|p_2(x)) = \mathcal{D}_{\text{KL}}((g_\sharp p_1)(x)\|(g_\sharp p_2)(x)). \tag{43}$$

Together,

$$\mathcal{G}(g_\sharp p) = \mathcal{G}(p). \tag{44}$$

$\square$

We now turn to the **proof of Proposition 2**:

*Proof.* We aim to find the affine-linear coupling layer $f_{\text{cpl}}$ minimizing $\mathcal{S}(\Sigma_1)$. By Lemma 1, $\mathcal{G}$ does not change.

The affine-linear coupling $f_{\text{cpl}}$ has the following form:

$$x_1 = \begin{pmatrix} \text{Diag}(r) & 0 \\ T & \text{Diag}(s) \end{pmatrix} \begin{pmatrix} p_0 \\ a_0 \end{pmatrix} + \begin{pmatrix} u \\ t \end{pmatrix} =: A x_0 + b. \tag{45}$$

To make the coupling affine-linear, $r, s \in \mathbb{R}_+^{D/2}$ are positive vectors, $u, t \in \mathbb{R}^{D/2}$ are vectors and $T \in \mathbb{R}^{D/2 \times D/2}$ is the matrix describing the linear dependence of $a_1$ on $p_0$.

By linearity of expectation, the mean of $x_1$ reads:

$$m_1 = A m_0 + b. \tag{46}$$

Write $S := \text{Diag}(s)$ and $R := \text{Diag}(r)$ so that the covariance of $x_1$ is given by:

$$\Sigma_1 := \text{Cov}[x_1] = A \Sigma_0 A^{\text{T}} \tag{47a}$$

$$= \begin{pmatrix} R & 0 \\ T & S \end{pmatrix} \begin{pmatrix} \Sigma_{0,pp} & \Sigma_{0,pa} \\ \Sigma_{0,ap} & \Sigma_{0,aa} \end{pmatrix} \begin{pmatrix} R & T^{\text{T}} \\ 0 & S \end{pmatrix} \tag{47b}$$

$$= \begin{pmatrix} R\Sigma_{0,pp}R & R(\Sigma_{0,pa}S + \Sigma_{0,pp}T^{\text{T}}) \\ (T\Sigma_{0,pp} + S\Sigma_{0,ap})R & (T\Sigma_{0,pa} + S\Sigma_{0,aa})S + (T\Sigma_{0,pp} + S\Sigma_{0,ap})T^{\text{T}} \end{pmatrix} \tag{47c}$$

Together, the non-Standardness of $x_1$ is given from Equation (8)

$$\mathcal{S}(m_1, \Sigma_1) \tag{48a}$$

$$= \tfrac{1}{2}\Big( \|m_1\|^2 + \text{tr}\,\Sigma_1 - D - \log \det \Sigma_1 \Big) \tag{48b}$$

$$= \tfrac{1}{2}\Big( \|Am_0 + b\|^2 + \text{tr}(R^2\Sigma_{0,pp}) + \text{tr}(T\Sigma_{0,pa}S) + \text{tr}(S^2\Sigma_{0,aa}) + \text{tr}(T\Sigma_{0,pp}T^{\text{T}}) \tag{48c}$$

$$\quad + \text{tr}(S\Sigma_{0,ap}T^{\text{T}}) - D - \log \det \Sigma_0 - \log \det R - \log \det S \Big). \tag{48d}$$

To find the minimum of $\mathcal{S}(m_1, \Sigma_1)$, minimize the above over $r, s, T$ and $b$:

$$\arg\min_{r,s,T,b} \mathcal{S}(m_1, \Sigma_1). \tag{49a}$$

It is easy to see that $b = -Am_0$ minimizes Equation (48) as in this case $m_1 = 0$.

At the minimum, we find for $r$:

$$0 = \frac{\partial \mathcal{S}(m_1, \Sigma_1)}{\partial r_m} = -\frac{1}{r_m} + r_m (\Sigma_{0,pp})_{mm}, \tag{50}$$

for some $m = 1, \ldots, D/2$. We read off that $r_m = (\Sigma_{0,pp})_{mm}^{-1/2}$. In matrix notation:

$$R = \text{Diag}(\Sigma_{0,pp})^{-1/2}. \tag{51}$$

For $s, T$, we find the system:

$$0 = \frac{\partial \mathcal{S}(m_1, \Sigma_1)}{\partial s_n} = -\frac{1}{s_n} + \sum_{j=1}^{D/2} T_{nj}(\Sigma_{0,pa})_{jn} + s_n(\Sigma_{0,ap})_{nn} \tag{52a}$$

$$0 = \frac{\partial \mathcal{S}(m_1, \Sigma_1)}{\partial T_{op}} = s_p(\Sigma_{0,pa})_{po} + \sum_{k=1}^{D/2} T_{pk}(\Sigma_{0,aa})_{ko}. \tag{52b}$$

Multiplying the first equation by $s_n$, we find in matrix notation:

$$I = \text{Diag}(T\Sigma_{0,pa}S + S^2\Sigma_{0,aa}) \tag{53a}$$

$$0 = S\Sigma_{0,pa} + T\Sigma_{0,pp}. \tag{53b}$$

We solve the second equation for $T$ (we use that $\Sigma_{0,pp}$ is invertible as it is positive definite):

$$T = -S\Sigma_{0,ap}\Sigma_{0,pp}^{-1}, \tag{54}$$

and insert into the first:

$$I = \mathrm{Diag}(-S\Sigma_{0,ap}\Sigma_{0,pp}^{-1}\Sigma_{0,pa}S + S^2\Sigma_{0,aa}) \tag{55a}$$

$$= \mathrm{Diag}(-S^2\Sigma_{0,ap}\Sigma_{0,pp}^{-1}\Sigma_{0,pa} + S^2\Sigma_{0,aa}). \tag{55b}$$

The last step is due to $\mathrm{Diag}(\cdot)$ linear and $S$ diagonal. We read off that:

$$S = \mathrm{Diag}(\Sigma_{0,aa} - \Sigma_{0,ap}\Sigma_{0,pp}^{-1}\Sigma_{0,pa})^{-1/2}. \tag{56}$$

Alternative solutions with negative signs are discarded by convention (without an effect on the covariance).

Inserting into Equation (47c), we find:

$$\Sigma_{1,pp} = R\Sigma_{0,pp}R = \mathrm{Diag}(\Sigma_{0,pp})^{-1/2}\Sigma_{0,pp}\,\mathrm{Diag}(\Sigma_{0,pp})^{-1/2} = M(\Sigma_{0,pp}), \tag{57a}$$

$$\Sigma_{1,pa} = R(\Sigma_{0,pa}S + \Sigma_{0,pp}T^{\mathrm{T}}) = R(\Sigma_{0,pa}S - \Sigma_{0,pp}\Sigma_{0,pp}^{-1}\Sigma_{0,pa}S) = 0, \tag{57b}$$

$$\Sigma_{1,ap} = \Sigma_{1,pa}^{\mathrm{T}} = 0, \tag{57c}$$

$$\Sigma_{1,aa} = (T\Sigma_{0,pa} + S\Sigma_{0,aa})S + (T\Sigma_{0,pp} + S\Sigma_{0,ap})T^{\mathrm{T}} \tag{57d}$$

$$= (-S\Sigma_{0,ap}\Sigma_{0,pp}^{-1}\Sigma_{0,pa} + S\Sigma_{0,aa})S + (S\Sigma_{0,ap}\Sigma_{0,pp}^{-1}\Sigma_{0,pp} - S\Sigma_{0,ap})\Sigma_{0,pp}^{-1}\Sigma_{0,pa}S \tag{57e}$$

$$= S(\Sigma_{0,aa} - \Sigma_{0,ap}\Sigma_{0,pp}^{-1}\Sigma_{0,pa})S = M(\Sigma_{0,aa} - \Sigma_{0,ap}\Sigma_{0,pp}^{-1}\Sigma_{0,pa}). \tag{57f}$$

This concludes the proof:

$$m_1 = 0, \qquad \Sigma_1 = \begin{pmatrix} M(\Sigma_{0,pp}) & 0 \\ 0 & M(\Sigma_{0,aa} - \Sigma_{ap}\Sigma_{pp}^{-1}\Sigma_{pa}) \end{pmatrix}. \tag{58}$$

$\square$

## B.3 Proof of Theorem 1

The following statement will help us along the way:

**Lemma 2.** *For $A \in \mathbb{C}^D$, $Q \in \{O(D), U(D)\}$ with the corresponding Haar measure $p(Q)$:*

$$\mathbb{E}_{Q\in p(Q)}[(QAQ^*)_{ii}] = \frac{1}{D}\,\mathrm{tr}(A). \tag{59}$$

*Proof.* By symmetry, $\mathbb{E}_Q[(QAQ^*)_{11}] = \mathbb{E}_Q[(QAQ^*)_{ii}]$ for $i = 1, \ldots, D$. Thus, $\mathbb{E}_Q[(QAQ^*)_{11}] = \frac{1}{D}\sum_{i=1}^D \mathbb{E}_Q[(QAQ^*)_{ii}] = \frac{1}{D}\mathbb{E}_Q[\mathrm{tr}(QAQ^*)] = \frac{1}{D}\,\mathrm{tr}\,A$. $\square$

When we write $Q^*$, we mean conjugate transpose if $Q$ is sampled from the unitary group $U(D)$, and transpose if $Q$ is from the orthogonal group $O(D)$. Whenever we only consider orthogonal $Q$, we will resort back to writing $Q^{\mathrm{T}}$.

This allows us to directly estimate $\mathbb{E}_{Q\sim p(Q)}[\log\det M_p^2]$:

**Lemma 3.** *With the definitions in Section 5.1, $p(Q)$ either the Haar measure of orthogonal or unitary matrices, and Assumption 2. Then:*

$$\mathbb{E}_{Q\sim p(Q)}[\log\det M_p^2] \geq 0. \tag{60}$$

*Proof.* $M_p$ is given by:

$$M_p^2 = \mathrm{Diag}(\Sigma_{0,pp})^{-1}. \tag{61}$$

The corresponding expectation value can be estimated via Jensen's inequality:

$$\mathbb{E}_{Q\sim p(Q)}[\log \det M_p^2] = \mathbb{E}_{Q\sim p(Q)}[\log \det \mathrm{Diag}(\Sigma_{0,pp})^{-1}] \tag{62a}$$

$$= -\mathbb{E}_{Q\sim p(Q)}[\log \prod_{i=1}^{D/2} (\Sigma_{0,pp})_{ii}] = -\sum_{i=1}^{D/2} \mathbb{E}_{Q\sim p(Q)}[\log(\Sigma_{0,pp})_{ii}] \tag{62b}$$

$$\geq -\sum_{i=1}^{D/2} \log \mathbb{E}_{Q\sim p(Q)}[(\Sigma_{0,pp})_{ii}] = -\frac{D}{2} \log \mathrm{tr}\, \Sigma/D \tag{62c}$$

$$= 0. \tag{62d}$$

By Assumption 2, $\mathrm{tr}\, \Sigma = D$. We have used Lemma 2 for evaluating $\mathbb{E}_{Q\sim p(Q)}[(\Sigma_{0,pp})_{ii}]$. $\qquad\square$

As mentioned in Section 5.1, the main difficulty in estimating $\mathbb{E}_{Q\sim p(Q)}[\mathcal{S}(\Sigma_1(Q))]$ lies in $\mathbb{E}_{Q\sim p(Q)}[\log \det M_a^2]$. The following subsections show a path to do so.

### B.3.1 Problem reformulation

In a first step, we reformulate this expectation so that it can be computed with the help of projected orbital measures [40].

We split the expectation over the Haar measure $p(Q)$ in two parts: One that defines which eigenvalues the $(D/2) \times (D/2)$ block $\Sigma_{0,aa}$ has (denote this as $Q_{ap}$) and, conditioned on this, another which rotates $\Sigma_{0,aa}$ into all possibles bases (denote this as $Q_a$). Formally, write $Q$ as:

$$Q = \begin{pmatrix} I & 0 \\ 0 & Q_a \end{pmatrix} Q_{ap}. \tag{63}$$

We will replace the Schur complement $\Sigma_{0,aa} - \Sigma_{0,ap}\Sigma_{0,pp}^{-1}\Sigma_{0,pa}$ appearing in Proposition 2 by the corresponding block of the precision matrix $P_0 := \Sigma_0^{-1} = (Q\Sigma^{-1}Q^*)^{-1} = Q\Sigma^{-1}Q^*$ (e.g. [39, Section (0.7.3)]):

$$(P_{0,aa})^{-1} = ((\Sigma_0^{-1})_{aa})^{-1} = \Sigma_{0,aa} - \Sigma_{0,ap}\Sigma_{0,pp}^{-1}\Sigma_{0,pa}. \tag{64}$$

We give more details in the proof of the following lemma, which formalizes this step:

**Lemma 4.** *Given the definitions in Section 5.1 and Assumption 2. It holds that*

$$\mathbb{E}_{Q\sim p(Q)}[\log \det M_a^2] \geq -\sum_{i=1}^{D/2} \log \mathbb{E}_{Q_a\sim p(Q_a|Q_{ap})}[((P_{0,aa})^{-1})_{ii}]. \tag{65}$$

*Proof.* By Proposition 2, $M_a^2$ is given by:

$$M_a^2 = \mathrm{Diag}(\Sigma_{0,aa} - \Sigma_{0,ap}\Sigma_{0,pp}^{-1}\Sigma_{0,pa})^{-1}. \tag{66}$$

Being diagonal, its determinant is given by the product of its diagonal entries:

$$\mathbb{E}_{Q\sim p(Q)}[\log \det M_a^2] = \mathbb{E}_{Q\sim p(Q)}[\log \prod_{i=1}^{D/2} (\Sigma_{0,aa} - \Sigma_{0,ap}\Sigma_{0,pp}^{-1}\Sigma_{0,pa})_{ii}^{-1}] \tag{67a}$$

$$= \sum_{i=1}^{D/2} \mathbb{E}_{Q\sim p(Q)}[\log((\Sigma_{0,aa} - \Sigma_{0,ap}\Sigma_{0,pp}^{-1}\Sigma_{0,pa})_{ii}^{-1})] \tag{67b}$$

$$= -\sum_{i=1}^{D/2} \mathbb{E}_{Q\sim p(Q)}[\log((\Sigma_{0,aa} - \Sigma_{0,ap}\Sigma_{0,pp}^{-1}\Sigma_{0,pa})_{ii})]. \tag{67c}$$

Evaluating this expression is hard mainly because the $\Sigma_{0,ap}\Sigma_{0,pp}^{-1}\Sigma_{0,pa}$ involves the inverse of $\Sigma_{0,pp} = (Q\Sigma Q^*)_{pp}$, which depends on $Q$.

To circumvent this, note the following property of any nonsingular matrix $M$ [39, Section (0.7.3)]. Split $M$ into blocks as:

$$M = \begin{pmatrix} A & B \\ B^* & C \end{pmatrix}, \tag{68}$$

and do the same for its inverse:

$$M^{-1} = \begin{pmatrix} A' & B' \\ B'^* & C' \end{pmatrix} \tag{69}$$

Then, $(A')^{-1} = A - BC^{-1}B^*$, which is called the Schur complement $M/C$. This means we can rewrite

$$\Sigma_{0,aa} - \Sigma_{0,ap}\Sigma_{0,pp}^{-1}\Sigma_{0,pa} = (P_{0,aa})^{-1}, \tag{70}$$

where $P_0 = \Sigma_0^{-1}$ is the *precision matrix* of the rotated data. Given a rotation $Q$, it can easily be obtained from the precision matrix of the data in its original rotation:

$$\Sigma_0 = Q\Sigma Q^*, \qquad P_0 = Q\Sigma^{-1}Q^*. \tag{71}$$

Inserting this, we find the expectation value:

$$\mathbb{E}_{Q \sim p(Q)}[\log \det M_a^2] = -\sum_{i=1}^{D/2} \mathbb{E}_{Q \sim p(Q)}[\log(((P_{0,aa})^{-1})_{ii})]. \tag{72}$$

The logarithm can be drawn out via Jensen's inequality:

$$-\sum_{i=1}^{D/2} \mathbb{E}_{Q \sim p(Q)}[\log(((P_{0,aa})^{-1})_{ii})] \geq -\sum_{i=1}^{D/2} \log(\mathbb{E}_{Q \sim p(Q)}[((P_{0,aa})^{-1})_{ii}]). \tag{73}$$

This concludes the statement. $\qquad\square$

### B.3.2 Projected orbit expectation

The theory of projected orbital measures describes the distribution of eigenvalues of a randomly projected submatrix of some given matrix. Let us formalize this:

Fix a diagonal matrix $A = \text{Diag}(a_1, \ldots, a_N)$. Then, then the *orbit* of $A$ is defined as:

$$\mathcal{O}_A := \{QAQ^* : Q \in U(D)\}. \tag{74}$$

(The same definition also exists for orthogonal $Q \in O(D)$, but we keep it to the level we require here). All matrices in the orbit of $\mathcal{O}_A$ have the same eigenvalues.

The natural measure (probability distribution) on the orbit $\mathcal{O}_A$ is given by the image of the Haar measure on the unitary group $U(D)$. This can be thought of as the uniform measure on the group of unitary rotations. We call this measure the *orbital measure*.

We now cut out the $K \times K$ top left corner out of every matrix in $\mathcal{O}_A$:

$$P_K \mathcal{O}_A := \{P_K Y : Y \in \mathcal{O}_A\}. \tag{75}$$

We call this the *projected orbit*. The matrix $P_K$ projects a matrix to its upper left corner:

$$P_K = (I_K; 0_{K \times (N-K)}). \tag{76}$$

The distribution of matrices in the projected orbit $P_K \mathcal{O}_A$ induced by the orbital measure is denoted as the *projected orbital measure* $\mu_{A,K}$. We are now interested in the eigenvalues of matrices in the projected orbit $P_K \mathcal{O}_A$.

Let $\text{spectrum}$ be the function that assigns a matrix $Y \in \mathbb{C}^{K \times K}$ its eigenvalues $y_1, \ldots, y_K$. We will make use of a result that gives the distribution of eigenvalues of matrices in the projected orbit $P_K \mathcal{O}_A$. This is called the *radial part of the projected orbital measure* and is denoted as $\nu_{A,K}(x_1, \ldots, x_K)$.

$$\nu_{A,K}(x_1, \ldots, x_K) = \mathbb{P}_{X \sim \mu_{A,K}}[\text{spectrum}(X) = (x_1, \ldots, x_K)]. \tag{77}$$

In other words, $\nu_{A,K}(x_1, \ldots, x_K)$ gives the probability density that a random matrix from the projected orbit of $A$ has exactly eigenvalues $(x_1, \ldots, x_K)$. Its functional form was shown by [40]:

**Theorem 4** (Radial part of projected orbital measure [40]). *Fix $A = (a_1, \ldots, a_D)$ with $a_1 < \cdots < a_D$. For any $K = 1, \ldots, D - 1$, the density of eigenvalues of*

$$\nu_{A,K}(x_1, \ldots, x_K) = c_{D,K} \frac{V(x_1, \ldots, x_K) \det[M(a_j; x_i, \ldots, x_{D-K+i})]_{i,j=1}^K}{\prod_{j-i \geq D-K+1}(x_j - x_i)}. \tag{78}$$

*Here, the constant is given by:*

$$c_{D,K} = \prod_{i=1}^{K-1} \binom{D - K + i}{i}, \tag{79}$$

*and $M(a; y_1, \ldots, y_N)$ is the B-spline:*

$$M(a; y_1, \ldots, y_n) := (N - 1) \sum_{i:y_i > a} \frac{(y_i - a)^{n-2}}{\prod_{r:r \neq i}(y_i - y_r)}, \tag{80}$$

*and $V$ is the Vandermonde polynomial:*

$$V(y_1, \ldots, y_n) = \prod_{i<j}(y_j - y_i). \tag{81}$$

We will make use of the following variant of the Vandermonde determinant where all powers greater or equal to some $k$ are increased by one:

**Lemma 5.** *For all $n \in \mathbb{N}$, $k = 1, \ldots, n - 1$ and distinct $a_i, i = 1, \ldots, n$:*

$$\det \begin{pmatrix} 1 & \cdots & a_1^{k-1} & a_1^{k+1} & \cdots & a_1^n \\ \vdots & & \vdots & \vdots & & \vdots \\ 1 & \cdots & a_n^{k-1} & a_n^{k+1} & \cdots & a_n^n \end{pmatrix} = V(a_1, \ldots, a_n) e_{n-k}(a_1, \ldots, a_n). \tag{82}$$

*with the elementary symmetric polynomial $e_K$ given by Equation (17).*

**Lemma 6.** *Fix $A = (a_1, \ldots, a_N)$ with $a_1 < \cdots < a_N$. For any $K = 1, \ldots, N - 1$, it holds that:*

$$\mathbb{E}_{a_1, \ldots, a_K \sim \nu_{A,K}(x_1, \ldots, x_K)}[x_1^{-1} + \cdots + x_K^{-1}] \tag{83a}$$

$$= (N - K)(-1)^{N-K} \sum_{j=1}^N a_j^{N-K-1} \log(a_j) R(a_j; a_{\neq j}) e_{K-1}(a_{\neq j}). \tag{83b}$$

*Here, $R$ is defined in Equation (17).*

*Proof.* We use the Andreief identity in the form of [44, Lemma 2.1]

$$\mathbb{E}_{x_1, \ldots, x_k \sim \nu_{A,K}}[\tfrac{1}{x_1} + \cdots + \tfrac{1}{x_k}] \tag{84a}$$

$$= Z^{-1} \int_{(\mathbb{R}^K)_+} (\tfrac{1}{x_1} + \cdots + \tfrac{1}{x_k}) \det(x_i^{j-1}) \det(M(x_i; a_j, \ldots, a_{j+N-K})) \, \mathrm{d}x_1 \cdots \mathrm{d}x_k \tag{84b}$$

$$= Z^{-1} \sum_{k=1}^K \det \int_{\mathbb{R}} x^{-\delta_{jk}} x^{j-1} M(x; a_i, \ldots, a_{i+N-K}) \, \mathrm{d}x \tag{84c}$$

$$= Z^{-1} \det \int_{\mathbb{R}} x^{j-1-\delta_{j1}} M(x; a_i, \ldots, a_{i+N-K}) \, \mathrm{d}x \tag{84d}$$

$$= Z^{-1} \det \begin{cases} \mu_{-1}(a_i, \ldots, a_{i+N-K}) & j = 1 \\ \mu_{j-1}(a_i, \ldots, a_{i+N-K}) & j > 1 \end{cases} \tag{84e}$$

where $\mu_k(t_1, \ldots, t_n)$ the $k$th moment of the B-spline with knots $t_1, \ldots, t_n$:

$$\mu_k(t_1, \ldots, t_n) = \int x^k M(x; t_1, \ldots, t_n) \, \mathrm{d}x. \tag{85}$$

We can now make use of the Hermite–Genocchi formula [41, Proposition 6.3]:

$$\int f^{(n-1)}(x) M(x; t_1, \ldots, t_n) \, dx = (n-1)! f[t_1, \ldots, t_n], \tag{86}$$

so we can rewrite

$$\mu_k(t_1, \ldots, t_n) = f_k[t_1, \ldots, t_n], \tag{87}$$

with

$$f_{-1}(x) = (n-1) x^{n-2} \log x, \tag{88a}$$

$$f_k(x) = \binom{n+k-1}{k}^{-1} x^{n+k-1}. \tag{88b}$$

Together, we find

$$\mathbb{E}_{x_1, \ldots, x_k \sim \nu_{A,K}} \left[ \tfrac{1}{x_1} + \cdots + \tfrac{1}{x_k} \right] = Z^{-1} \det(f_{i-1-\delta_{1i}}[a_j, \ldots, a_{j+N-K}]). \tag{89}$$

The right hand side can be identified with the right hand side of [41, Proposition 6.4]. It is equal to:

$$Z^{-1} \det(f_i[a_j, \ldots, a_{j+N-K}]) \tag{90a}$$

$$= Z^{-1} \left( \prod_{0 < j-i \le N-K} (a_j - a_i)^{-1} \right) \begin{vmatrix} 1 & \cdots & 1 \\ a_1 & \cdots & a_N \\ \vdots & & \vdots \\ a_1^{N-K-1} & \cdots & a_N^{N-K-1} \\ f_1(a_1) & \cdots & f_1(a_N) \\ \vdots & & \vdots \\ f_K(a_1) & \cdots & f_K(a_N) \end{vmatrix} \tag{90b}$$

$$= Z^{-1} \left( \prod_{0 < j-i \le N-K} (a_j - a_i)^{-1} \prod_{k=1}^{K} \binom{N-K+k}{k}^{-1} \right) (N-K) \begin{vmatrix} 1 & \cdots & 1 \\ a_1 & \cdots & a_N \\ \vdots & & \vdots \\ a_1^{N-K-1} & \cdots & a_N^{N-K-1} \\ a_1^{N-K-1} \log a_1 & \cdots & a_N^{N-K-1} \log a_N \\ a_1^{N-K+1} & \cdots & a_N^{N-K+1} \\ \vdots & & \vdots \\ a_1^{N-1} & \cdots & a_N^{N-1} \end{vmatrix} \tag{90c}$$

$$=: C_2 \det(M_{ij}) \tag{90d}$$

Here, $C_2$ reduces to:

$$C_2 = \frac{N-K}{V(a_1, \ldots a_n)}. \tag{91a}$$

Then, the determinant of $M_{ij}$ reads:

$$\det M_{ij} = \sum_{j=1}^{N} (-1)^{N-K+1+j} a_j^{N-K-1} \log(a_j) V(a_{\ne j}) \sum_{\substack{i_1 < \cdots < i_{K-1} \\ i_{\ldots} \ne j}} a_{i_1} \cdots a_{i_{K-1}} \tag{92a}$$

$$= V(a)(-1)^{N-K} \sum_j a_j^{N-K-1} \log(a_j) R(a_j; a_{\ne j}) e_{K-1}(a_{\ne j}), \tag{92b}$$

where $R(a_j; a_{\ne j})$ collects all the terms in $V(a)$ that were not contained in $V(a_{\ne j})$ up to sign:

$$R(a_j; a_{\ne j}) = \prod_{\substack{i=1 \\ i \ne j}}^{n} \frac{1}{a_i - a_j} = (-1)^{j-1} V(a_{\ne j}) / V(a). \tag{93}$$

Note that the sign of $R(a_j; a_{\neq j})$ flips from $j \to j+1$, so the alternating nature of the above series remains.

Together, the desired expectation value reads:

$$\mathbb{E}_{x_1,\ldots,x_k \sim \nu_{A,K}}[\tfrac{1}{x_1} + \cdots + \tfrac{1}{x_k}] \tag{94a}$$

$$= (N-K)(-1)^{N-K} \sum_{j=1}^{N} a_j^{N-K-1} \log(a_j) R(a_j; a_{\neq j}) e_{K-1}(a_{\neq j}), \tag{94b}$$

which concludes the proof. $\qquad\square$

We now connect this result to our situation. This paves the path from the reformulation in Lemma 4 to Theorem 1.

**Corollary 1.** *For the definitions in Section 5.1 and when Assumptions 3 and 4 are fulfilled, it holds that:*

$$\mathbb{E}_{Q \sim p(Q)}[\log \det M_a^2] \geq \tfrac{D}{2} \log\left((-1)^{\frac{D}{2}+1} \sum_{i=1}^{D} \lambda_i^{1-\frac{D}{2}} \log(\lambda_i) R(\lambda_i^{-1}; \lambda_{\neq i}^{-1}) e_{\frac{D}{2}-1}(\lambda_{\neq i}^{-1})\right). \tag{95}$$

*Proof.* Then, Lemma 2 tells us how to integrate over $Q_a$.

$$\mathbb{E}_{Q_a \sim p(Q_a | Q_{ap})}[((P_{0,aa})^{-1})_{ii}] = \text{tr}((Q_{ap} P Q_{ap}^*)^{-1}) = \sum_{i=1}^{D/2} a_i(Q_{ap})^{-1}. \tag{96}$$

Here, we denote by $a_i(Q_{ap})$ the $i$th eigenvalue of $P_0 = Q_{ap} P Q_{ap}^*$, which depends on the "outer" rotation $Q_{ap}$.

We substitute the expectation over $Q_{ap}$ with an expectation over the projected eigenvalues of the rotated precision matrix $P_0$:

$$\mathbb{E}_{Q_{ap} \sim p(Q)}[\mathbb{E}_{Q_a \sim p(Q_a | Q_{ap})}[((P_{0,aa})^{-1})_{ii}] = \text{tr}((Q_{ap} P Q_{ap}^*)^{-1})] \tag{97a}$$

$$= \mathbb{E}_{a_1,\ldots,a_{D/2} \sim \nu_{A,D/2}(a_1,\ldots,a_{D/2} | \lambda_1^{-1},\ldots,\lambda_D^{-1})}[a_1^{-1} + \cdots + a_{D/2}^{-1}]. \tag{97b}$$

Here $X = (\lambda_1^{-1}, \ldots \lambda_D^{-1})$ contains the eigenvalues of the precision matrix $P$, the inverse of the covariance $\Sigma$. Lemma 6 with $K = D/2$ tells us how to evaluate the above expression. Insert the result into Lemma 4 to obtain the result. $\qquad\square$

### B.3.3 Summary

We can now collect the above pieces to build the **proof of Theorem 1**:

*Proof.* Equation (15) is the version of the non-Standardness after a single layer when Assumptions 1 and 2 are fulfilled. Insert Lemma 3 (passive part) and Corollary 1 (active part) to obtain the result. The former required Assumption 2 and the latter Assumptions 3 and 4 to hold. $\qquad\square$

### B.3.4 Handling of imaginary part

If we allow for unitary rotations $Q \in U(D)$, real-valued data is typically rotated into imaginary space. In fact, the case that the input remains real even has probability zero:

$$\mathbb{P}[Qx \in \mathbb{R}^D] = 0. \tag{98}$$

This does not pose a problem for our theory: The covariance matrix is positive definite also for complex data and so it has a positive determinant and trace, which are the only quantities entering the non-Standardness $\mathcal{S}$ (see Equation (8)).

### B.4 Proof of Theorem 2

**Lemma 7.** *With the definitions in Section 5.1 and $p(Q)$ the Haar measure over the orthogonal group $O(D)$:*

$$\mathbb{E}_{Q\sim p(Q)}[\log\det M_a^2] \geq D/2\log\left(1 - \frac{DD}{2(D+1)(D-1)}\frac{\mathrm{Var}[\lambda]}{\lambda_{\max}}\right). \tag{99}$$

*Proof.* By Proposition 2, $M_a^2$ is given by:

$$M_a^2 = \mathrm{Diag}(\Sigma_{0,aa} - \Sigma_{0,ap}\Sigma_{0,pp}^{-1}\Sigma_{0,pa})^{-1}. \tag{100}$$

The determinant of a diagonal matrix is equal to the product of the entries on the diagonal. By the permutation symmetry of $p(Q)$, we can pick the entry in the upper left corner:

$$\mathbb{E}_{Q\sim p(Q)}[\log\det M_a^2] = D/2\mathbb{E}_{Q\sim p(Q)}[\log((M_a^{-2})_{11})] \leq -D/2\log\mathbb{E}_{Q\sim p(Q)}[(M_a^2)_{11}]. \tag{101}$$

The last step is due to the Jensen inequality.

We are left with computing $\mathbb{E}_{Q\sim p(Q)}[(M_a^2)_{11}]$:

$$\mathbb{E}_{Q\sim p(Q)}[(M_a^2)_{11}] = \mathbb{E}_{Q\sim p(Q)}[(\Sigma_{0,aa} - \Sigma_{0,ap}\Sigma_{0,pp}^{-1}\Sigma_{0,pa})_{11}] \tag{102a}$$

$$= \mathbb{E}_{Q\sim p(Q)}[(\Sigma_{0,aa})_{11}] - \mathbb{E}_{Q\sim p(Q)}[(\Sigma_{0,ap}\Sigma_{0,pp}^{-1}\Sigma_{0,pa})_{11}] \tag{102b}$$

$$= \frac{1}{D}\mathrm{tr}\,\Sigma_0 - \mathbb{E}_{Q\sim p(Q)}[(\Sigma_{0,ap}\Sigma_{0,pp}^{-1}\Sigma_{0,pa})_{11}]. \tag{102c}$$

The first expectation can be exactly computed via Lemma 2.

The average trace of the second matrix is not so easy to evaluate. As $\Sigma_{0,pp}^{-1}$ is positive definite, we can replace it with the worst case in the expectation:

$$(\Sigma_{0,ap}\Sigma_{0,pp}^{-1}\Sigma_{0,pa})_{11} \geq (\Sigma_{0,ap}\lambda_{\max}^{-1}I\Sigma_{0,pa})_{11} = (\Sigma_{0,ap}\Sigma_{0,pa})_{11}\lambda_{\max}^{-1}. \tag{103}$$

$\lambda_{\max}$ is the largest eigenvalue of $\Sigma$, which does not depend on $Q$.

The expectation value can now be computed exactly:

$$\mathbb{E}_{Q\sim p(Q)}[(\Sigma_{0,ap}\Sigma_{0,pa})_{11}] = \sum_{i=1}^{D/2}\mathbb{E}_{Q\sim p(Q)}[(\Sigma_{0,ap})_{1i}(\Sigma_{0,pa})_{i1}] \tag{104a}$$

$$= D/2\mathbb{E}_{Q\sim p(Q)}[(\Sigma_{0,ap})_{11}^2]. \tag{104b}$$

The last step is because each summand will have the same contribution. Writing the matrix multiplication out explicitly:

$$(\Sigma_{0,ap})_{11}^2 = (Q\,\mathrm{Diag}(\lambda_1,\ldots,\lambda_D)Q^*)_{11}^2 = \left(\sum_{j=1}^{D}Q_{1j}\lambda_j Q_{(D/2+1)j}\right)^2 \tag{105}$$

Again by symmetry, we can exchange axes and write 2 instead of $D/2+1$ in what follows:

$$\left(\sum_{j=1}^{D}Q_{1j}\lambda_j Q_{(D/2+1)j}\right)^2 = \left(\sum_{j=1}^{D}Q_{1j}\lambda_j Q_{2j}\right)^2 = \sum_{j,k=1}^{D}\lambda_j\lambda_k Q_{1j}Q_{2j}Q_{1k}Q_{2k}. \tag{106}$$

Taking the expectation, we use the linearity of the expectation and are left with the following monomials of elements of $Q$:

1. $j = k$: $\mathbb{E}_{Q\sim p(Q)}[Q_{1j}^2 Q_{2j}^2] = \mathbb{E}_{Q\sim p(Q)}[Q_{11}^2 Q_{21}^2]$ as we can exchanges axes,

2. $j \neq k$: $\mathbb{E}_{Q\sim p(Q)}[Q_{1j}Q_{2j}Q_{1k}Q_{2k}] = \mathbb{E}_{Q\sim p(Q)}[Q_{11}Q_{21}Q_{12}Q_{22}]$ as we can exchange axes.

By [42], these amount to the following integrals of monomials of entries of random orthogonal matrices and the corresponding values:

$$1. \ \langle 2 \quad 2 \rangle = \frac{1}{D(D+2)}, \tag{107a}$$

$$2. \ \left\langle \begin{matrix} 1 & 1 \\ 1 & 1 \end{matrix} \right\rangle = -\frac{1}{D(D-1)(D+2)}. \tag{107b}$$

Together, we find

$$\mathbb{E}_{Q\sim p(Q)}[(M_a^2)_{11}] = 1 - \frac{1}{2(D+2)\lambda_{\max}} \left( \sum_{j=1}^{D} \lambda_j^2 - \frac{1}{D-1} \sum_{j\neq k} \lambda_j \lambda_k \right) \tag{108a}$$

$$= 1 - \frac{D^2}{2(D-1)(D+2)} \frac{\mathrm{Var}[\lambda]}{\lambda_{\max}}. \tag{108b}$$

Here, $\mathrm{Var}[\lambda] = \frac{1}{D} \operatorname{tr} \Sigma^2 - (\frac{1}{D} \operatorname{tr} \Sigma)^2$ is the variance of the eigenvalues of $\Sigma$.

Insert this to obtain the result. $\qquad\square$

**Lemma 8.** *With the definitions in Section 5.1:*

$$\mathbb{E}_{Q\sim p(Q)}[\log \det M_a^2] \geq D/2 \log\left(1 - \frac{DD}{2(D+1)(D-1)} \frac{\mathrm{Var}[\lambda]}{\lambda_{\max}}\right). \tag{109}$$

*Proof.* The idea is to lower bound

$$\frac{\mathrm{Var}[\lambda]}{\lambda_{\max}} \tag{110}$$

by some function of $L$. We make use of following arithmetic mean-geometric mean (AM-GM) inequality by [43]:

$$\frac{\mathrm{Var}[\lambda]}{2\lambda_{\max}} \leq \bar{\lambda} - g \leq \frac{\mathrm{Var}[\lambda]}{2\lambda_{\min}}, \tag{111}$$

where $g$ is the geometric mean of the eigenvalues:

$$g := \left( \prod_{i=1}^{D} \lambda_i \right)^{1/D}. \tag{112}$$

We can write the loss $L$ directly via $g$ and vice versa:

$$L = -\frac{1}{2} \log g^D = -\frac{D}{2} \log g, \tag{113a}$$

$$g = \exp(-2L/D). \tag{113b}$$

Rewrite Equation (111) to our needs:

$$\frac{\mathrm{Var}[\lambda]}{\lambda_{\max}} = \frac{\mathrm{Var}[\lambda]\lambda_{\min}}{\lambda_{\max}\lambda_{\min}} = \frac{2}{\kappa} \frac{\mathrm{Var}[\lambda]}{2\lambda_{\min}} \geq \frac{2}{\kappa}(1-g), \tag{114}$$

with $\kappa$ the condition number of the covariance $\Sigma$.

As we want a bound that merely depends on the loss, we upper bound $\kappa$ using a function of the loss, yielding a lower bound on $\mathrm{Var}[\lambda]/\lambda_{\max}$ that merely depends on the loss. The maximum of the condition value is given by:

$$\max_{\substack{\lambda_1,\ldots,\lambda_D \\ \sum_i \lambda_i = D \\ \prod_i \lambda_i^{1/D} = g}} \kappa = \frac{1 + \sqrt{1 - g^D}}{1 - \sqrt{1 - g^D}}. \tag{115}$$

This yields the required lower bound:

$$\frac{\mathrm{Var}[\lambda]}{\lambda_{\max}} \geq 2 \frac{1 - \sqrt{1 - g^D}}{1 + \sqrt{1 - g^D}}(1-g), \tag{116}$$

which results in an overall upper bound:

$$\mathbb{E}_{Q \in O(D)}[\mathcal{S}(\Sigma_1(Q))] \leq \mathcal{S}(\Sigma) + \frac{D}{4} \log\left(1 - \frac{D^2}{(D-1)(D+2)} \frac{1 - \sqrt{1 - g^D}}{1 + \sqrt{1 - g^D}}(1 - g)\right). \quad (117)$$

Replacing the expression in Equation (113b) for $g$ yields the statement. $\square$

We summarize to obtain the **proof of Theorem 2**:

*Proof.* Equation (15) is the form of non-Standardness $\mathcal{S}(\Sigma_1)$ (see Equation (8)) we need to evaluate when Assumptions 1 and 2 hold. Into this equation, insert Lemma 3 together with Lemma 7 for the first bound. For the second bound, insert Lemmas 3 and 8. $\square$

### B.5 Proof of Theorem 3

*Proof.* The non-Standardness will not increase by the action of a single layer given in Proposition 2 (compare Equation (14)). This holds regardless of the rotations of the individual blocks $Q_1, \ldots Q_L$, so $\mathcal{S}(\Sigma) = \mathcal{S}(\Sigma_0) \geq \mathcal{S}(\Sigma_1) \geq \cdots \geq \mathcal{S}(\Sigma_L)$. It is easy to see that $\gamma$ decreases as $\mathcal{S}$ decreases by using $\mathcal{S} > 0$ to check that

$$\frac{\partial \gamma}{\partial \mathcal{S}} > 0. \quad (118)$$

Together, we have:

$$\gamma\big(\mathcal{S}(\Sigma_{L-1})\big) \leq \cdots \leq \gamma\big(\mathcal{S}(\Sigma_0)\big). \quad (119)$$

Rewrite Theorem 2 as follows:

$$\mathbb{E}_{Q \in O(D)}[\mathcal{S}(\Sigma_1(Q))] \leq \gamma\big(\mathcal{S}(\Sigma_0)\big)\mathcal{S}(\Sigma), \quad (120)$$

and apply repeatedly:

$$\mathbb{E}_{Q_1,\ldots,Q_L \in O(D)}[\mathcal{S}(\Sigma_L)] \leq \mathbb{E}_{Q_1,\ldots,Q_{L-1} \in O(D)}[\gamma(\mathcal{S}(\Sigma_{L-1}))\mathcal{S}(\Sigma_{L-1})] \quad (121a)$$
$$\leq \gamma(\mathcal{S}(\Sigma))\mathbb{E}_{Q_1,\ldots,Q_{L-1} \in O(D)}[\mathcal{S}(\Sigma_{L-1})] \quad (121b)$$
$$\leq \cdots \leq \gamma(\mathcal{S}(\Sigma))^L \mathcal{S}(\Sigma_0) \quad (121c)$$

This shows the statement. $\square$

## C  Compatible coupling architectures

All statements in this paper apply to the following architectures, where we assume each layer to be equipped with ActNorm [6]. To shorten the notation, we consider how a single dimension is transformed and rewrite the dependence on $p_0$ via a parameter vector $\theta = \theta(p_0)$, which is usually computed by a feed-forward neural network:

$$y = c(x; \theta), \quad (122)$$

short for:

$$(a_1)_i = c\big((a_0)_i; \theta_i(p_0)\big). \quad (123)$$

- **Affine coupling flows** in the form of NICE [4], RealNVP [5] and GLOW [6]:

$$c(x; \theta) = sx + t. \quad (124)$$

  Here, $\theta = [s; t] \in \mathbb{R}_+ \times \mathbb{R}$.
- **Nonlinear squared flow** [19]:

$$c(x; \theta) = ax + b + \frac{c}{1 + (dx + h)^2}, \quad (125)$$

  for $\theta = [a, b, c, d, h] \in \mathbb{R}^5$.

- **SOS polynomial flows** [27]:

$$c(x;\theta) = \int_0^x \sum_{\kappa=1}^k \left( \sum_{l=0}^r a_{l,\kappa} u^l \right)^2 \mathrm{d}u + t. \tag{126}$$

Here, $\theta = [t; (a_{l,\kappa})_{l,\kappa}] \in \mathbb{R} \times \mathbb{R}^{rk}$.

- **Flow++** [18]:

$$c(x;\theta) = s\sigma^{-1} \left( \sum_{j=1}^K \pi_j \sigma \left( \frac{x - \mu_j}{\sigma_j} \right) \right) + t. \tag{127}$$

Here, $\theta = [s; t; (\pi_j, \mu_j, \sigma_j)_{j=1}^K] \in \mathbb{R}_+ \times \mathbb{R} \times (\mathbb{R} \times \mathbb{R} \times \mathbb{R}_+)^K$ and $\sigma$ is the logistic function.

- **Spline flows** in the form of piecewise-linear, monotone quadratic [20], cubic [21], and rational quadratic [22] splines. Here, $c$ is a spline of the corresponding type, parameterized by knots $\theta$.

- **Neural autoregressive flow** [26] parameterize $c(x;\theta)$ by a feed-forward neural network, which can be shown to be bijective if all weights are positive and all activation functions are strictly monotone.

  One can also restrict the neural network $c(x;\theta)$ to have positive output and integrate it numerically. This was introduced as **unconstrained monotonic neural networks** [24].