# OpenReview forum: "Whitening Convergence Rate of Coupling-based Normalizing Flows"
_NeurIPS.cc/2022/Conference — NeurIPS 2022 Accept_

### Official Review · Reviewer_X9Dq · 2022-06-15

**Rating:** 7
**Confidence:** 3
**Soundness:** 3 good
**Presentation:** 3 good
**Contribution:** 3 good

**Summary:**

In the article, the authors analyze the convergence of affine coupling flows. So far, it has only been shown that this type of normalizing flows converges weakly to an arbitrary target distribution. However, from this result it cannot be concluded that it converges in maximum likelihood.

The authors obtain a stronger convergence guarantee by decomposing the KL-divergence into a part measuring the non-Gaussianity, i.e. the KL between the flow density and a Gaussian with mean and variance as the data, and the non-Standardness, i.e. the KL-divergence of the just mentioned Gaussian distribution and a Standard Normal distribution. They prove bounds of the non-Standardness for single and multiple affine coupling layers and thereby prove convergence while quantifying the convergence rate. The authors validate the bound and the convergence rate empirically on EMNIST as well as a toy dataset.

**Questions:**

* Where did you show that the non-Gaussianity is not increasing?
* Could similar results be derived for other flow architectures?
* On the right plot of Figure 2, it looks like that deviations are almost exclusively present in two quadrants, which might hint towards a systematic deviation. How do the authors explain this phenomenon?

**Limitations:**

The main limitation is that the authors only proved and verified the convergence rate of the non-Standardness, but not the non-Gaussianity.

**Strengths And Weaknesses:**

**Strengths**

Affine Coupling Flows are very popular, being used in countless scientific research projects as well as applications. Hence, getting a better understanding of whether and how they converge is a very important problem. The authors give an intuitive decomposition of the training objective, and I appreciate its visualization in Figure 1. They very thoroughly analyze the non-Standardness, provide and prove several bounds, thereby proving and quantifying convergence. The proofs in the appendix are very detailed and I tried to verify them. To the best of my knowledge, they seem correct but I am not an expert regarding the convergence of machine learning algorithms, so I might have missed something.

I like the short paragraph before the conclusion section very much since the authors provide an intuitive heuristic derived from their results, i.e. that the non-Standardness is reduced by about 50% with every affine coupling layer added to a flow model. This will be very useful for practitioners.

**Weaknesses**

The main weakness of the paper is that it only considers bounds for the non-Standardness. They only show empirically for EMNIST that the non-Gaussianity decreases similarly; however, this might be highly dependent on the nature of the target distribution, a fact not being reflected in the paper. The authors claim on lines 146-147 that it does not increase but it is not clear to me where this is shown. It might be buried somewhere in the appendix, but this result should be placed more prominently in the main paper.

The paper focuses on affine coupling flows. However, it would also be interesting to briefly discuss whether the bounds could be extended to other flow architectures.

Since the authors computed the non-Standardness to verify their bound, I was wondering whether the non-Standardness could not be reduced by just whitening the data through a linear transformation before training a flow. In this case, only the non-Gaussianity needs to be minimized. I suppose that this is impractical for large high-dimensional datasets; however, the authors could try this on a smaller toy dataset.

**Conclusion**

I think this is a strong paper and I lean towards accepting it and, hence, I score this paper with a 6. If the authors can clarify some of the criticism I mentioned, I am willing to increase my score.

**Edit**

Given that the authors addressed my concerns and even showed that their theorems extend to all coupling flows I will upgrade my score to 7.

---

> ### Author Response · Authors · 2022-08-02
> **More details on non-Gaussianity, other architectures, and the deviations in Figure 2**
>
> We cordially thank you for your helpful feedback and hope to address the limitations you mentioned in the following:
>
> > Where did you show that the non-Gaussianity is not increasing?
>
> We have updated Section 4 to make this more prominent (see Proposition 2 and surrounding paragraphs). G does not change because the employed coupling layer is linear (see Lemma 1 in Appendix B). However, the fact that G is constant was an understatement: The proof of Proposition 2 allows a broader class of couplings that achieve exactly the covariance given in Eq. (10), but also decrease G. In other words: *While our theory only shows how S can be brought to zero, it leaves room for the involved coupling layers to reduce G*.
> Showing a convergence rate for G is beyond the scope of this work, however.
>
> > Could similar results be derived for other flow architectures?
>
> Thank you for pointing this out. Indeed, our result holds for arbitrary coupling flows (i.e. architectures that split incoming vectors in halves and leave one half unchanged) equipped with ActNorm (for r and u to be present). This is the case for NICE (Dinh et al. 2014), RealNVP (Dinh et al. 2016), and GLOW (Kingma & Dhariwal, 2018); Flow++ (Ho et al. 2019); nonlinear-squared flow (Ziegler & Rush 2019); linear, quadratic (Müller et al. 2019), cubic (Durkan et al. 2019a), and rational quadratic splines (Durkan et al. 2019b); neural autoregressive flows (Huang et al. 2020), and unconstrained monotonic neural networks (Wehenkel & Louppe, 2019). *For all these architectures, our theory guarantees the demonstrated convergence of the non-Standardness*. Note that none of our theorems or proofs have to be altered to be applied here.
>
> The updated version of our paper reflects this insight and we propose to rename the paper to “Whitening Convergence Rate of Coupling Flows”.
>
> > On the right plot of Figure 2, it looks like that deviations are almost exclusively present in two quadrants, which might hint towards a systematic deviation. How do the authors explain this phenomenon?
>
> We attribute this phenomenon to the training dynamics (batch size 2048 is almost two orders of magnitude less than the number of parameters to be learnt ((D/2)^2 + D=154,448). We observed that increasing the batch size or reducing the learning rate makes the deviations vanish to arbitrarily small values.

---

> > ### Comment · Reviewer_X9Dq · 2022-08-09
> > **Upgrade of my score**
> >
> > I thank the authors for addressing my concerns. I upgraded my score to 7.

---

### Official Review · Reviewer_XD8h · 2022-07-03

**Rating:** 7
**Confidence:** 3
**Soundness:** 3 good
**Presentation:** 3 good
**Contribution:** 3 good

**Summary:**

A **normalizing flow** (NF) is a generative model that transforms a complex distribution using a chain of invertible transformations into a simple distribution. Furthermore, a NF models the exact likelihood using the change of variables formula and can be trained via the maximum likelihood principle.

Many different transformations have been considered in the past. One of those is the so-called affine coupling layer. **This paper is about** a theoretical analysis of the whitening convergence rate of affine coupling flows. To do so, the authors show that the KL divergence used for training can be split into two parts handling "non-Gaussianity" and "non-Standardness". The authors then provide bounds on the "non-Standardness" and argue that each affine coupling layer (followed by a rotation (for reasons)) typically reduces the "non-Standardness" by approx. 50%.

**Questions:**

**Q1:** To me, the use of $r$ and $u$ in Equation (4) appear to be non-standard within the literature and I assume they are learned as parameters during training. If those are not used (i.e. the standard affine coupling is being used) I assume the bounds do no longer hold. How large is the influence of having those? Can it be at least empirically shown that the same convergence is achieved?

**Q2:** Would you agree that instead of learning 2D additional parameters per layer ($r,d$), one could learn the mean and a diagonal (or even the full covariance) of the Gaussian base distribution. Such a distribution is still easy to handle but would change the setting: the flow would just have to handle the (more challenging) part of "non-Gaussianity". Moreover, the analysis presented only considers bounds on the "non-Standardness" part of the loss, however, the "non-Gaussianity" part seems to be the real challenge. If "non-Standardness" decreases, "non-Gaussianity" could still remain a problem, no?

**Q3:** What are the assumptions on the functions $s(\cdot)$ and $t(\cdot)$? Line 105 states they can be arbitrary but I would assume that their expressiveness largely influences the amount of layers/blocks needed.


**Limitations:**

The authors flagged that they discussed the negative societal impact of their work in line 410, however, I do not see such a discussion except the comment in lines 410-412.


**Strengths And Weaknesses:**

**Strengths:**
The paper is well written and mostly easy to follow. The authors begin by arguing that maximum likelihood estimation is equivalent to minimizing the KL divergence and decompose this KL divergence into two parts handling "non-Gaussianity" and "non-Standardness". Then, they derive bounds for the "non-Standardness" part for a single affine coupling layer as well as for a concatenation of such. Although I did not check every detail of the proofs, the derivation appears to be sound.

**Weaknesses:**
- The use of $r$ and $u$ in Equation (4) appear to be non-standard within the literature. It is unclear if the results still hold if this *architecture change* is not present. (See also Question 1 below.)
- If the loss can be seen as "non-Gaussianity" plus "non-Standardness", the main challenge seems to be the "non-Gaussianity", not "non-Standardness". However, the paper is only concerned with bounding "non-Standardness". The more important part seems to be ignored. Hence, the usefulness of the bounds is questionable. (See also Question 2 below.)

**Minor comments/questions:**
- The pdf should be built without boxes around links
- Why is the $x$-axis scaled non-linearly in Figure 1?
- Please rephrase lines 89-91.
- The actual (negative log-)likelihood objective could be stated in Section 3 along with a case of a flow that consists of multiple transformations/layers for completeness.
- Please avoid starting a sentence with variables/symbols (e.g., lines 91 & 106).
- lines 96, 127, etc.: (1) might be mistaken as a reference to an equation, hence use (i).
- lines 99, 115: RealNVP does not mention any rotation. A rotation layer is used in Glow [6], however, the rotation is learned and not fixed. Nevertheless, I think Glow should be mentioned there.
- line 103: The passive part is not transformed!
- Equation (4): Please introduce $p_0$ and $a_0$, e.g., within the sentence above.
- Lines 116-118 should be mentioned earlier.
- Cite or provide a link for FrEIA.
- line 181: The comma should be after Thm. 2.
- line 291: "We iterate adding such"
- line 302: additive
- [4,5] are ICLR papers.
- Why are arXiv identifiers provided, i.e., in [3,8,...]?

**After the author feedback:**
I have read the author feedback and the other reviews. Especially reviewer X9Dq shared my concern that only the non-Standardness part of the loss is considered whereas the non-Gaussianity part seems to be the real challenge. However, the authors provided clarifications, actually to both points I raised in **Weaknesses** above, which is why is raise my score to 7.

---

> ### Author Response · Authors · 2022-08-02
> **More details on architecture, learning the base distribution, non-Gaussianity, and assumptions on s and t**
>
> We cordially thank you for your helpful feedback and hope to address the limitations you mentioned in the following:
>
> > Q1: How large is the influence of having [$r$ and $u$]? Can it be at least empirically shown that the same convergence is achieved? Can it be at least empirically shown that the same convergence is achieved?
>
> Practically, $r$ and $u$ have been observed to be helpful. They were introduced in Glow as ActNorm as a replacement for an invertible BatchNorm (Kingma & Dhariwal, NeurIPS 2018). We have updated the paper to make this more clear.
>
> Technically, the single-layer bounds still hold, as we estimate the effect of r and u on the passive dimensions to be zero. The multi-layer bounds require the use of r and u for the Assumption 2 to persist (i.e. tr Sigma = D). Empirically, the same convergence behavior is achieved.
>
> > Q2: Would you agree that [...] one could learn the mean, [...] diagonal (or even the full covariance) of the Gaussian base distribution?
>
> The mean and diagonal of the Gaussian base distribution are already implicitly learnt via ActNorm (see Q1). Regarding the off-diagonal entries of the covariance, one would need to parametrize the covariance in a clever way such that (i) it remains positive definite, and (ii) we can easily compute its determinant. While the issue (i) can be addressed in various ways, we are not aware of parameterizations that also satisfy (ii), in addition to (i).
>
> Our work implies that it is not even necessary to modify the base distribution, as we show that the non-Standardness can be reduced using just a few layers.
>
> > Q2 (continued): If "non-Standardness" decreases, "non-Gaussianity" could still remain a problem, no?
>
> We have made Proposition 2 and the text around it more precise to make the relation to non-Gaussianity more clear: We guaranteed that G never increases by the coupling which minimizes non-Standardness S. This is, in fact, an understatement: The proof of Proposition 2 allows a broader class of couplings that achieve exactly the covariance given in Eq. (10), but also decrease G. In other words: *While our theory only shows how S can be brought to zero, it leaves room for the involved coupling layers to reduce G*. We give the details for this more general result in the updated text surrounding Proposition 2.
> Showing a convergence rate for G is beyond the scope of this work, however.
>
> This also brings to light that our theory does not only hold for affine coupling blocks (i.e. Glow/RealNVP). Indeed, our results also apply to all other coupling architectures that can represent linear functions. This is the case for all coupling architectures aware to us, i.e. NICE (Dinh et al. 2014), RealNVP (Dinh et al. 2016), and GLOW (Kingma & Dhariwal, 2018); Flow++ (Ho et al. 2019); nonlinear-squared flow (Ziegler & Rush 2019); linear, quadratic (Müller et al. 2019), cubic (Durkan et al. 2019a), and rational quadratic splines (Durkan et al. 2019b); neural autoregressive flows (Huang et al. 2020), and unconstrained monotonic neural networks (Wehenkel & Louppe, 2019). *For all these architectures, our theory guarantees the demonstrated convergence of the non-Standardness*. Note that none of our theorems or proofs have to be altered to be applied here. The updated version of our paper reflects this insight and we propose to rename the paper to “Whitening Convergence Rate of Coupling Flows”. We thank reviewer X9Dq for inspiring this generalization.
>
> > Q3: What are the assumptions on the functions s(⋅) and t(⋅)?
>
> In general, we would expect that s and t need to be able to represent complicated functions to learn arbitrary distributions p(x).
> However, our theory only requires that s can represent a constant value, and t a linear function in p to reduce the non-Standardness as presented. In regards to non-Gaussianity, more flexible functions will be needed, as linear couplings cannot change it (see Q2). The exact condition will depend on the coupling architecture used.
>
> Many thanks for the other detailed comments. We have already included some of your suggestions and will work on the remainder for the final version of this paper. Regarding the log-scaling of the network depth, few layers are sufficient to fit the non-Standardness, so we focus the figure on the first layers. We still show the later layers in the plot to show that the architecture is well-equipped to learn the dataset.

---

> > ### Comment · Reviewer_XD8h · 2022-08-09
> > **Updated Review**
> >
> > Thank you very much for your answers and clarifications. I updated my review and raised the score.

---

### Official Review · Reviewer_HZCb · 2022-07-11

**Rating:** 6
**Confidence:** 3
**Soundness:** 3 good
**Presentation:** 3 good
**Contribution:** 2 fair

**Summary:**

This paper studies normalizing flows, which is an approach for generative modeling relying on minimizing the KL divergence between a data distribution fed through a pushforward map implemented with neural networks and the standard Gaussian distribution. To generate samples, one can sample from the standard Gaussian and then invert this network. The paper studies a particular variant of normalizing flows called affine coupling flows, which use RealNVP layers. The main theoretical result analyzes a decomposed KL divergence into a non-Gaussian part and a nonstandardness part. Theorem 1 shows that the nonstandardness can decrease after the application of an affine coupling block, and the result is made interpretable in Theorem 2. Theorem 3 then shows that for deeper networks the nonstandardness can decrease at an exponential rate. Experiments confirm the upper bound and show the exponential decrease with respect to deep networks.

**Questions:**

- Are you able to write your bounds as high probability bounds rather than bounds on the expected decrease?
- What proof techniques used might be more broadly interesting and applicable?

**Limitations:**

The limitations of the analysis do not appear to be discussed in the text, contrary to the fact that the authors mention that they are discussed in Section 6.

**Strengths And Weaknesses:**

Strengths
- There is not existing theory studying the optimization properties of RealNVP layers. This paper tackles an important open question evaluating theoretically the effectiveness of these layers.
- The paper is generally well-written.
- The decomposition of the KL divergence into parts seems to be a useful tool for the analysis of these layers.

Weaknesses
- The result stands alone and is not close to a complete result. What I mean by this is that the non-Gaussianity term and the non-standardness results are optimized together when the RealNVP layers are implemented. This means that, in theory, the actual minimization procedure may not decrease the non-standardness at all, since there may be a tradeoff between these terms. Therefore, studying one of these terms alone seems unrealistic, and I don't know how it should relate to practical methods.
- While the exponential decrease is interesting and seems to hold in practice, I don't see how this theoretical result could be useful in guiding methodology. The authors suggest a heuristic for the reduction of the nonstandardness but this only seems to be motivated by the experiments and not the theory. Can be we guaranteed that the reduction is close to 50% for real data, or can we estimate somehow the expected reduction from a dataset?
- The tight upper bound takes exponential time to compute and is hard to interpret, while the looser bounds appear to be quite loose according to the plots.

Update:
I still have reservations about the interaction of the non-Gaussianity term with the non-standardness term. However, the author's answers alleviate at least some of my concern, and so I will raise my score.

---

> ### Author Response · Authors · 2022-08-02
> **More details on non-Gaussianity, 50%-heuristic, high probability bounds, proof techniques**
>
> We cordially thank you for your helpful feedback and hope to address the limitations you mentioned in the following:
>
> > the actual minimization procedure may not decrease the non-standardness at all, since there may be a tradeoff between these terms. Therefore, studying one of these terms alone seems unrealistic, and I don't know how it should relate to practical methods.
>
> It is true that the actual minimization of the joint loss L = G + S may result in a different coupling than the one proposed in our paper. However, the layer we propose can always be achieved and we show that it causes a fast convergence of the non-Standardness.
>
> Regarding non-Gaussianity: The coupling that we propose in Proposition 2 was built to reduce non-Standardness, but it has freedom to also reduce non-Gaussianity. We adapted the explanation in the paper to make this more precise: Proposition 2 merely guaranteed that G never increases by the coupling which minimizes non-Standardness S. This was, in fact, an understatement: The proof allows a broader class of couplings that achieve exactly the covariance given in Eq. (10), but also decrease G. In other words: *While our theory only shows how S can be brought to zero, it leaves room for the involved coupling layers to reduce G*. We give the details in the updated text around Proposition 2. Showing a convergence rate for G is beyond the scope of this work, however.
>
> We hope that this also clarifies the limitations of our work, mentioned in the last section.
>
> In terms of practical implications, our main takeaway is that we don’t need to worry about reducing the non-Standardness, as already few layers suffice to reduce it by large.
>
> The clarification about non-Gaussianity also brings to light that our theory does not only hold for affine coupling blocks (i.e. Glow/RealNVP). Our results also apply to all other coupling architectures that can represent linear functions. This is the case for all coupling architectures aware to us, i.e. NICE (Dinh et al. 2014), RealNVP (Dinh et al. 2016), and GLOW (Kingma & Dhariwal, 2018); Flow++ (Ho et al. 2019); nonlinear-squared flow (Ziegler & Rush 2019); linear, quadratic (Müller et al. 2019), cubic (Durkan et al. 2019a), and rational quadratic splines (Durkan et al. 2019b); neural autoregressive flows (Huang et al. 2020), and unconstrained monotonic neural networks (Wehenkel & Louppe, 2019). *For all these architectures, our theory guarantees the demonstrated convergence of the non-Standardness*. Note that none of our theorems or proofs have to be altered to be applied here. The updated version of our paper reflects this insight and we propose to rename the paper to “Whitening Convergence Rate of Coupling Flows”. We thank reviewer X9Dq for inspiring this generalization.
>
> > Can be we guaranteed that the reduction is close to 50% for real data, or can we estimate somehow the expected reduction from a dataset?
>
> The ‘close to 50%’-heuristic is backed by our theory in the limit where S is close to zero (depending on the dimension, we obtain between 50%-55% reduction). For all other cases, one can evaluate the precise bound on; we conjecture that the tight bound computed for the initial data will also hold for several layers.
>
> > Are you able to write your bounds as high probability bounds rather than bounds on the expected decrease?
>
> This is a relevant point, thank you for pointing it out. We strongly suspect a concentration result to hold, as we are dealing with high dimensional random (rotation) matrices. This is backed by our experiments, where increasing dimensions quickly made the variance of the involved expectations vanish (see e.g. the small Interquartile Range in Figure 3). Showing such a concentration result is technically challenging and therefore beyond the scope of this work.
>
> > What proof techniques used might be more broadly interesting and applicable?
>
> We identify two new main ideas: (i) reduce the loss by a single layer and draw conclusions for a deep flow; (ii) split the KL divergence into constituents via a Pythagorean theorem.

---

### Official Review · Reviewer_r6j2 · 2022-07-11

**Rating:** 7
**Confidence:** 3
**Soundness:** 4 excellent
**Presentation:** 4 excellent
**Contribution:** 3 good

**Summary:**

This paper provides new theoretical insights on the affine coupling in normalizing flows.
The contributions of the paper are clearly presented within 4 points at the end of the introduction section.

**Questions:**

We find this work very interesting. We do not have any question.

**Limitations:**

yes

**Strengths And Weaknesses:**

The paper is well written.

The contributions are theoretical contributions that allow to better understand the underlying mechanism of the affine coupling in normalizing flows.

It is worth noting that many theoretical results in normalizing flows were derived for the optimal transport. This paper investigates affine coupling by providing novel insights. This includes providing a novel point of view of the affine coupling layers as whitening transformation, and convergence analysis with explicit convergence rate and deep network guarantee.

---

> ### Author Response · Authors · 2022-08-02
> **Thank you for your feedback.**
>
> If any questions arose since your review, we are happy to address them.
>
> Please note that we clarified two points in the paper: (i) We give more details on the non-Gaussianity. (ii) We show that our results apply not only to affine coupling flows, but to all coupling architectures known to us. See the general comment "More details on non-Gaussianity and generalization to all coupling architectures" for more details.

---

### Author Response · Authors · 2022-08-02
**More details on non-Gaussianity and generalization to all coupling architectures**

We cordially thank all reviewers for their helpful feedback. We have updated the paper in two ways:

1) We give more details on the consequences of our results for non-Gaussianity G. Proposition 2 already guaranteed that G never increases by the coupling which minimizes non-Standardness S. This was, in fact, an understatement: The proof of Proposition 2 allows a broader class of couplings that achieve exactly the covariance given in Eq. (10), but also decrease G. In other words: *While our theory only shows how S can be brought to zero, it leaves room for the involved coupling layers to reduce G.* We give the details in the updated text around Proposition 2. Showing a convergence rate for G is beyond the scope of this work, however.

2) This clarification also brings to light that our theory does not only hold for affine coupling blocks (i.e. Glow/RealNVP). Our results also apply to all other coupling architectures that can represent linear functions. This is the case for all coupling architectures aware to us:
NICE (Dinh et al. 2014), RealNVP (Dinh et al. 2016), and GLOW (Kingma & Dhariwal, 2018); Flow++ (Ho et al. 2019); nonlinear-squared flow (Ziegler & Rush 2019); linear, quadratic (Müller et al. 2019), cubic (Durkan et al. 2019a), and rational quadratic splines (Durkan et al. 2019b); neural autoregressive flows (Huang et al. 2020), and unconstrained monotonic neural networks (Wehenkel & Louppe, 2019).
*For all these architectures, our theory guarantees the demonstrated convergence of the non-Standardness.* Note that none of our theorems or proofs have to be altered to be applied here. The updated version of our paper reflects this insight and we propose to rename the paper to “Whitening Convergence Rate of Coupling Flows”.
We thank reviewer X9Dq for inspiring this generalization.

The supplementary material is now included in the main pdf for better readability.

---

### Meta-Review · Area_Chair_Rbzy · 2022-08-20

**Recommendation:** Accept
**Confidence:** Certain

**Metareview:**

In this work, the authors analyze the convergence of affine coupling flows by providing a theoretical analysis of the whitening convergence rate. While previous analyses were derived for the optimal transport, the reviewers have appreciated the point of view provided by viewing the affine coupling layers as whitening transformations. They have however regretted that the non-Gaussianity term was ignored in the theoretical analysis. All the reviewers found the work relevant, interesting, with meaningful theoretical and empirical results. Therefore I do recommend acceptance of this paper.

**Award:**

No

---

### Decision · Program_Chairs · 2022-09-14

Accept